# VFLAIR: A Research Library and Benchmark for Vertical Federated Learning

**Tianyuan Zou**[1], **Zixuan Gu**[2], **Yu He**[3], **Hideaki Takahashi**[4], **Yang Liu**[*1,5], and **Ya-Qin Zhang**[1]

[1]Institute for AI Industry Research, Tsinghua University, Beijing, China
[2]Weiyang College, Tsinghua University, Beijing, China
[3]School of Computer Science, Fudan University, Shanghai, China
[4]College of Arts and Sciences, The University of Tokyo, Tokyo, Japan
[5]Shanghai Artificial Intelligence Laboratory, China

## Abstract

Vertical Federated Learning (VFL) has emerged as a collaborative training paradigm that allows participants with different features of the same group of users to accomplish cooperative training without exposing their raw data or model parameters. VFL has gained significant attention for its research potential and real-world applications in recent years, but still faces substantial challenges, such as in defending various kinds of data inference and backdoor attacks. Moreover, most of existing VFL projects are industry-facing and not easily used for keeping track of the current research progress. To address this need, we present an extensible and lightweight VFL framework `VFLAIR` (available at `https://github.com/FLAIR-THU/VFLAIR`), which supports VFL training with a variety of models, datasets and protocols, along with standardized modules for comprehensive evaluations of attacks and defense strategies. We also benchmark 11 attacks and 8 defenses performance under different communication and model partition settings and draw concrete insights and recommendations on the choice of defense strategies for different practical VFL deployment scenarios.

## 1 Introduction

The concept of Federated Learning (FL) was first introduced by Google in 2016 (McMahan et al., 2016a) describing a cross-device scenario where millions of mobile users collaboratively train a shared model using their local private data without centralizing these data. This scenario is regarded as Horizontal FL (HFL) (Yang et al., 2019b) as data are partitioned by sample. In another type of FL, regarded as Vertical FL (VFL) (Yang et al., 2019b), data are partitioned by feature. VFL is often applied in industrial collaborative learning scenarios where each organization controls disjoint features of a common group of users. In VFL, local data and local model are kept private at each participant. Instead, local model outputs and their gradients are transmitted between parties.

VFL has drawn increasing attention from both academic and industry in recent years with hundreds of research papers published every year and a number of open-sourced projects released, including FATE (FedAI-maintainers; Liu et al., 2021c), Fedlearner (ByteDance), PaddleFL (Baidu), Pysyft (Ryffel et al., 2018; Romanini et al., 2021), FedTree (Li et al., 2022c), and FedML (He et al., 2020). Real-world industrial cases are also emerged in the field of advertising (Cai, 2020; Lin, 2021) and finance (Cheng et al., 2020; 2022) etc. However, mainstream VFL projects such as FATE are industrial grade and not designed for keeping up with research advances.

Meanwhile, research interests for VFL have been growing rapidly over the past years, focusing on improving various aspects of VFL protocols, such as communication efficiency (Fu et al., 2021; Liu et al., 2022b; Castiglia et al., 2022b; Fu et al., 2022c), robustness to attacks (Liu et al., 2021b; Cheng et al., 2021; Li et al., 2022b; Zou et al., 2022; 2023; Sun et al., 2022; Yang et al., 2022a), model utility (Li et al., 2022d; Yang et al., 2022b; Feng &Yu, 2020; Feng et al., 2022), and fair incentive designs (Liu et al., 2021a; Qi et al., 2022). For communication efficiency, methods like decrease

---

[*]Corresponds to Yang Liu (liuy03@air.tsinghua.edu.cn).

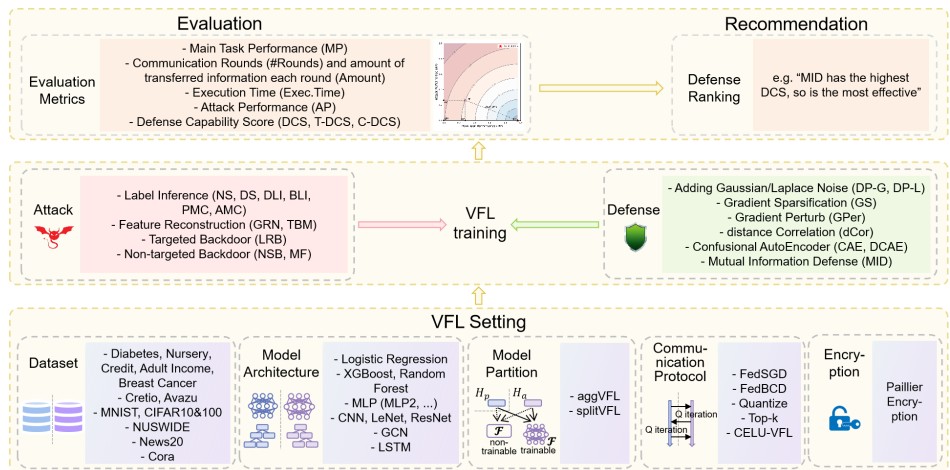

Figure 1: An overview of the Components of VFLAIR.

communication rounds using multiple local updates between each round (Liu et al., 2022b; Fu et al., 2022c) or compress information (Castiglia et al., 2022b) have been proposed. As for data security and privacy, various attacks injected by one or multiple parties aiming to either steal other parties' private label (Li et al., 2022b; Fu et al., 2022a; Zou et al., 2022), private features (Jin et al., 2021; Luo et al., 2021; Li et al., 2022a; Jiang et al., 2022a; Ye et al., 2022b), sensitive attributes (Song &Shmatikov, 2020) and sample relations (Qiu et al., 2022), or negatively impact the model behavior (Liu et al., 2021b; Zou et al., 2022) have been put forward. Multiple defending methods have also been proposed to tackle these threats, including adding noise (Dwork, 2006; Zou et al., 2022; Li et al., 2022b), sparsifying gradients (Aji &Heafield, 2017; Fu et al., 2022a; Zou et al., 2022), discreting gradients (Fu et al., 2022a), label differential privacy (Ghazi et al., 2021; Yang et al., 2022a), adding distance correlation regularizor (Sun et al., 2022; Vepakomma et al., 2019), disguising labels (Zou et al., 2022), adding mutual information regularizer (Zou et al., 2023), adversarial training (Sun et al., 2021b; Li et al., 2022a) or performing robust feature recovery (Liu et al., 2021b). However, each of these defenses are evaluated under specific tasks and settings, lacking of key insights and metrics on evaluating these defense strategies to defend all possible attacks in practical deployment.

To facilitate future research for VFL, we introduce a lightweight and comprehensive VFL framework, namely `VFLAIR`, which includes not only basic VFL training and inference for a variety of models and settings but also efficiency enhancement techniques and multiple defense methods that mitigate potential threats. Moreover, we perform extensive experiments on combinations of the above settings using multiple datasets to provide different perspectives on VFL efficiency and safety. We believe `VFLAIR` and these benchmark results will provide researchers with useful tools and guidance for their future work. Our contributions are summarized in the following:

**(1).** We design `VFLAIR`, a lightweight and extensible VFL framework that aims to facilitate research development of VFL (see Fig. 1). We design standardized pipelines for VFL training and validation, supporting 13 datasets, 29 different local model architectures including linear regression, tree and neural networks, 6 different global models, 2 model partition settings, 5 communication protocols, 1 encryption method, 11 attacks and 8 defense methods, each implemented as a distinct module and can be easily extended.

**(2).** We propose new evaluation metrics and modules, and perform extensive experiments to benchmark various perspectives of VFL, from which we draw key insights on VFL system design choice, in order to promote future development and practical deployment of VFL.

## 2 RELATED WORK

A number of open-source FL projects have been developed supporting VFL. FATE (FedAI-maintainers; Liu et al., 2021c) is an industry-grade FL project which supports a variety of model architectures and secure computation protocols; Fedlearner (ByteDance) is specialized in advertising scenarios; PaddleFL (Baidu) supports 2-party and 3-party VFL with MPC protection; Pysyft (Ryffel et al., 2018; Romanini et al., 2021) introduces PyVertical, which focus on SplitNN-type of VFL

settings; FedTree (Li et al., 2022c) focuses on tree-based VFL only; FedML (He et al., 2020) supports basic training of VFL with logistic regression models. Real-world industrial applications have been witnessed in domains such as advertising (Cai, 2020; Lin, 2021) and finance (Cheng et al., 2020; 2022). These works demonstrate the widespread interest and the practical significance of VFL. However, these works are often relatively heavy-weight as they are designed for industrial deployment. On the other hand, most existing benchmarks on FL focus on HFL scenario (Chai et al., 2020; Lai et al., 2022; Zhang et al., 2023). For VFL, (Kang et al., 2022) evaluates several defense strategies for data reconstruction attacks; SLPerf (Zhou et al., 2023) focuses on benchmarking and comparing various kinds of splitNN scenarios like splitVFL. No existing work provides a comprehensive evaluation covering a variety of key aspects of VFL settings, including model performance, communication efficiency and robustness to attacks. Due to space limitation, we discuss works on VFL definition and emerging fields of research interest in Appendix A.

## 3 VFL FRAMEWORK

In a typical VFL setting with $K$ parties, each party owns their local private feature $\{X_k\}_{k=1}^K$ and local model $\{G_k\}_{k=1}^K$ with parameters $\{\theta_k\}_{k=1}^K$ respectively. Only one party controls the private label information $Y$ and is referred to as *active* party while other parties are referred to as *passive* parties. The active party also controls a global trainable model parameterized by $\varphi$ (splitVFL) or global non-trainable function $F$ (aggVFL) to aggregate each party's local model output. Note in tree-based VFL the global function is an aggregation function that identifies the optimal feature split based on feature splitting information received from all parties. Without loss of generality, we assume that the $K^{th}$ party is the active party while other $K - 1$ parties are passive parties.

In the collaborative training process of NN-based VFL, each party computes its local feature embedding $H_k = G_k(X_k, \theta_k), k = 1, \ldots, K$. The active party collects $\{H_k\}_{k=1}^K$ and gets the final prediction $\hat{Y} = F(H_1, \ldots, H_K, \varphi)$. The loss $\mathcal{L} = \ell(Y, \hat{Y})$ is calculated at the active party. The gradient w.r.s. to $H_k$ as $g_k = \frac{\partial \mathcal{L}}{\partial H_k}, k = 1, \ldots, K$ are then calculated and transmitted back to each party by the active party. Using these gradients, each party performs local model updates by SGD using $\nabla_{\theta_k} \mathcal{L} = \frac{\partial \mathcal{L}}{\partial \theta_k} = \frac{\partial \mathcal{L}}{\partial H_k} \frac{\partial H_k}{\partial \theta_k}, k = 1, \ldots, K$. Also the active party performs model update with SGD on global model $F$ if it is trainable using $\nabla_{\varphi} \mathcal{L} = \frac{\partial \mathcal{L}}{\partial \varphi}$. In the inference procedure, the same is done but without the backward gradient descent to get the prediction of labels. If the exchange of $H_k$ and $g_k$ is performed each round, such VFL protocol is referred as FedSGD protocol. On the other hand, if communication is done every $Q > 1$ steps of local updates, such protocol is referred to as FedBCD (Liu et al., 2022b). The training procedure is shown in detail in Algorithm 1 in Appendix B. Training and inference procedures of tree-based VFL are included in Algorithm 3 in Appendix B.

## 4 OVERVIEW OF VFLAIR

**Implemented Components.** An overview of the components of VFLAIR is shown in Fig. 1. VFLAIR incorporates not only basic VFL training and testing process for both NN-based and tree-based VFL of various settings, but also multiple existing efficiency enhancement techniques, data leakage and model utility impairing attacks as well as defending methods that aim to mitigate potential threats. VFLAIR provides support for both aggVFL and splitVFL with easily adjustable model architectures. Currently, VFLAIR supports 5 communication protocols to improve communication efficiency. Also, 11 existing attacks and 8 defenses are supported. Moreover, VFLAIR supports the comprehensive assessment of defense performance using carefully designed metrics (see Sec. 5.3), based on which defense strategy recommendations can be provided. Paillier Encryption (Cheng et al., 2021) is also supported to further protect transmitted results. In total, 13 datasets from a diverse range of industrial domains, including but not limited to medical, financial, and recommendation are supported. Datasets are partitioned into either balanced or unbalanced subsets to mimic real-world participants based on human knowledge (see Tab. 10 in Appendix H.1 for a full list of dataset and partition methods).

**How to use and extend.** VFLAIR is a light-weight and comprehensive VFL framework that can be launched on a single GPU or CPU (see Tab. 9 for its system requirement compared to FATE). VFLAIR facilitates the easy integration of different datasets for model training and inference through simple dataset loading and partitioning functions. New attacks and defenses can be quickly incorporated into the framework thanks to the modular structure. Step-by-step guidance for using and extending VFLAIR are included in Appendix C. The workflow of VFLAIR is also shown in Appendix D.

Table 1: Summary of attacks for NN-based VFL

| Attack Type | Attack | Requirements / Limitations | Attacker Party | Attack Performance (AP) |
|---|---|---|---|---|
| Label Inference (LI) | Norm-based Scoring (NS) (Li et al., 2022b) | binary classification, sample-level | passive | AUC of inferred labels |
| | Direction-based Scoring (DS) (Li et al., 2022b) | | | |
| | Direct Label Inference (DLI) (Li et al., 2022b; Zou et al., 2022) | sample-level | | ratio of correctly inferred labels |
| | Batch-level Label Inference (BLI) (Zou et al., 2022) | - | | |
| | Passive Model Completion (PMC) (Fu et al., 2022a) | auxiliary labeled data for each class | | |
| | Active Model Completion (AMC) (Fu et al., 2022a) | | | |
| Feature Reconstruction (FR) | Generative Regression Network (GRN) (Luo et al., 2021) | black-box | active | $1 - \text{MSE}(U_0, U_{rec})$ |
| | Training-based Back Mapping by model inversion (TBM) (Li et al., 2022a) | white-box, auxiliary i.i.d. data | | |
| Targeted Backdoor (TB) | Label Replacement Backdoor (LRB) (Zou et al., 2022) | $\geq 1$ sample of target class | passive | ratio of triggered samples inferred as target class |
| Non-targeted Backdoor (NTB) | Noisy-sample Backdoor (NSB) (Zou et al., 2023) | - | passive | MP difference between total and noisy/missing samples |
| | Missing Feature (MF) (Liu et al., 2021b) | - | | |

## 5 VFL Benchmark

### 5.1 VFL Settings, Models and Datasets

Using `VFLAIR`, We benchmark the VFL main task performance using 13 datasets including MNIST (Yann LeCun), CIFAR10 (Krizhevsky &Hinton, 2009), CIFAR100 (Krizhevsky &Hinton, 2009), NUSWIDE (Chua et al., 2009), Breast Cancer (Street et al., 1993), Diabetes (Kahn), Adult Income (Becker &Kohavi, 1996), Criteo (Guo et al., 2017), Avazu (Qu et al., 2018), Cora (McCallum et al., 2000), News20 (Lang, 1995),Credit (Dua &Graff, 2017) and Nursery (Dua &Graff, 2017). Detailed data partition strategies are included in Appendix H.1. We explore 2 distinct architectures, namely aggVFL and splitVFL, and comprehensively benchmark their performance. The local models used for both settings are detailed in Tab. 10 in Appendix H.1. For global model $F$, a global softmax function is applied under aggVFL setting while a 1-layer fully-connected model serves as the global model for splitVFL setting (except for Cora dataset, for which a 1-layer graph convolution layer is applied). Additionally, we investigate the impact of different communication protocols by comparing FedBCD (Liu et al., 2022b) ($Q = 5$) and CELU-VFL (Fu et al., 2022c) ($Q = 5, W = 5$), as well as compression mechanisms Quantize ($b = 16$) (Castiglia et al., 2022b) and Top-k ($r = 0.9$) (Castiglia et al., 2022b) to the conventional FedSGD, as discussed in Sec. 3 and further provide insights into the communication cost reduction achieved by communication efficient protocols, as well as the impact of FedBCD when various attacks and defenses are deployed. We also evaluate the impact of the number of participating parties as well as the type of local model (logistic regression, tree, NN) on the main task performance of VFL. For tee-based VFL, we further benchmark both Random Forrest and XGBoost algorithms. Moreover, for tree-based VFL, we employ Paillier Encryption (Cheng et al., 2021) to protect transmitted information and measure its impact on computation efficiency. Details of corresponding datasets and models for tree-based VFL can be found in Tab. 10 and Appendix H.1 in the appendix.

### 5.2 Attacks and Defenses

We benchmark the performance of 11 attacks with 8 defenses on 3 datasets including MNIST (Yann LeCun), CIFAR10 (Krizhevsky &Hinton, 2009) and NUSWIDE (Chua et al., 2009). For these evaluations, we mainly consider a VFL setting with 1 active party and 1 passive party following original works (Li et al., 2022b; Luo et al., 2021; Li et al., 2022a), denoted as party $a, p$ respectively, with each party owning their local feature $X_a, X_p$ and local model $G_a, G_p$. The local model output of the active and passive party are denoted as $H_a, H_p$ respectively. We summarized the evaluated attacks in Tab. 1. Note in Tab. 1, NS and DS attacks can only be applied to binary classification scenarios; "sample-level" indicates that the attack requires gradient information for each sample, whereas "batch-level" means only batch-level gradients information are available; "black-box" indicates that the model is kept private at the party under attack, but can be queried by the attacker and honestly return the output to the attacker, whereas "white-box" means the attacker has access to the model; $\text{MSE}(U_0, U_{rec}) = \mathbb{E}[(u_0^{(f)} - u_{rec}^{(f)})^2]$ where $u_0^{(f)}, u_{rec}^{(f)}$ are the $f^{th}$ feature of original input $U_0$ and recovered input $U_{rec}$ respectively. LI, FR and NTB attacks are inference time attacks that are launched separately from VFL training procedure while only TB attacks are training time attacks. Detailed descriptions and hyper-parameters of all the evaluated attacks are provided in Appendices F and H.3, mainly following the original papers. Defense methods are summarized in Tab. 2 with respective hyper-parameters. Detailed descriptions of defense methodologies and implementations are included in Appendices G and H.4.

Table 2: Summary of defense methods and tested hyper-parameter values for NN-based VFL.

| Defense | Methodology | Hyper-parameter | Hyper-parameter Values |
|---|---|---|---|
| G-DP (Dwork, 2006; Fu et al., 2022a; Zou et al., 2022) | add noise to gradients or local prediction | DP Strength | $0.0001, 0.001, 0.01, 0.1$ |
| L-DP (Dwork, 2006; Fu et al., 2022a; Zou et al., 2022) | add noise to gradients or local prediction | DP Strength | $0.0001, 0.001, 0.01, 0.1$ |
| GS (Aji &Heafield, 2017; Fu et al., 2022a; Zou et al., 2022) | drop gradient elements close to 0 | Sparsification Rate | $95.0\%, 97.0\%, 99.0\%, 99.5\%$ |
| GPer (Yang et al., 2022a) | perturb gradient with that of other class | Perturbation Strength | $0.0001, 0.001, 0.01, 0.1$ |
| dCor (Sun et al., 2022; Vepakomma et al., 2019) | distance correlation regularization | Regularizer Strength | $0.0001, 0.01, 0.1, 0.3$ |
| CAE (Zou et al., 2022) | disguise label | Confusion Strength $\lambda$ | $0.0, 0.1, 0.5, 1.0$ |
| DCAE (Zou et al., 2022) | discrete gradient in addition to CAE | Confusion Strength $\lambda$ | $0.0, 0.1, 0.5, 1.0$ |
| MID (Zou et al., 2023) | mutual information (MI) regularization | Regularizer Strength $\lambda$ | $0.0, 1e^{-8}, 1e^{-6}, 1e^{-4}, 0.01, 0.1, 1.0, 1e^2, 1e^4$ |

## 5.3 EVALUATION METRICS

**Main Task Performance (MP).** MP is defined as the final model prediction accuracy on the test dataset, which reveals the utility of the VFL system.

**Communication and Computation Efficiency.** Number of communication rounds (#Rounds) and the amount of data transferred for each round (Amount) are used for measuring communication efficiency. Execution Time (Exec.Time) is used to measure computation efficiency.

**Attack Performance (AP).** The definition of AP varies with respect to the type of the attack and is summarized in Tab. 1. Definition details can be seen in Appendix E.

**Defense Capability Score (DCS).** Intuitively, an ideal defense should not compromise the utility of the original main task and should thwart the attack completely. Therefore, considering that both AP and MP are key metrics to evaluate defenses. We further propose **Defense Capability Score (DCS)**, to directly compare all the defenses under one unified metric. Let $df = (\text{AP}, \text{MP})$ represents the performance of a defense on an AP-MP graph, then we define its defense capability score (DCS) based on the distance between $df$ to an ideal defense $df^* = (\text{AP}^*, \text{MP}^*)$. $\text{MP}^*$ is the MP of VFL without defense and $\text{AP}^*$ is set to $0.0$ representing the performance of a completely incapable attacker. Then, we formulate the definition of DCS as:

$$\text{DCS} = \frac{1}{1 + D(df, df^*)} = \frac{1}{1 + \sqrt{(1-\beta)(\text{AP} - \text{AP}^*)^2 + \beta(\text{MP} - \text{MP}^*)^2}}, \quad (1)$$

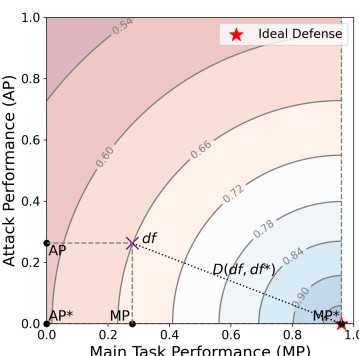

where $D(\cdot)$ is a user-defined distance function. Here we use Euclidean distance with an adjustable trade-off weighting parameter $\beta$. A visualization of DCS on an AP-MP graph with $\beta = 0.5$ can be seen in Fig. 2. A point closer to the bottom-right corner of an AP-MP graph has a higher DCS score indicating a better defense capability, consistent with intuition. $\beta = 0.5$ is used in our experiments.

**Type-level Defense Capability Score (T-DCS).** T-DCS is the DCS score averaged by attack type. Treating all $I_j$ attacks of the same attack type $j$ as equally important, we average DCS for each attack $i$ to get T-DCS for attack type $j$:

$$\text{T-DCS}_j = \frac{1}{I_j} \sum_{i=1}^{I_j} \text{DCS}_i. \quad (2)$$

Figure 2: A visual illustration example of DCS. The numbers on the contour lines are DCSs calculated with $\beta = 0.5$.

**Comprehensive Defense Capability Score (C-DCS).** C-DCS is a comprehensive assessment of the capability of a defense strategy with respect to all kinds of attacks and is a weighted average of T-DCS as shown in Eq. (3):

$$\text{C-DCS} = \sum_{j \in \mathcal{A}} w_j \text{T-DCS}_j, \text{ with } \sum_{j \in \mathcal{A}} w_j = 1.0. \quad (3)$$

Weights $\{w_j\}_{j \in \mathcal{A}}$ can be tailored to user preference. In our experiments, we simply use an unbiased weight $w_j = \frac{1}{|\mathcal{A}|}$ for each attack type $j \in \mathcal{A} = \{\text{LI}, \text{FR}, \text{TB}, \text{NTB}\}$.

## 6 EVALUATION RESULTS

### 6.1 VFL MAIN TASK PERFORMANCE

We first comprehensively evaluate the impact of various settings on the performance of VFL. Details on training model and training hyper-parameters for the following experiments are included in Tab. 10 (in Appendix H.1) and Appendix H.2 respectively.

Table 3: MP under 4 different settings of NN-based VFL. $Q = 5$ when FedBCD is applied. In "#Rounds" column, the first and second numbers are the communication rounds needed to reach the specified MP for FedSGD and FedBCD respectively.

| Dataset | aggVFL, FedSGD | aggVFL, FedBCD | #Rounds | splitVFL, FedSGD | splitVFL, FedBCD | #Rounds |
|---|---|---|---|---|---|---|
| MNIST | 0.972±0.001 | 0.971 ±0.001 | 150 / 113 | 0.973±0.001 | **0.974±0.001** | 180 / 143 |
| NUSWIDE | 0.887±0.001 | 0.882±0.001 | 60 / 26 | **0.888±0.001** | 0.884±0.001 | 60 / 29 |
| Breast Cancer | 0.914±0.033 | 0.919±0.029 | 5 / 3 | **0.925±0.028** | 0.907±0.045 | 5 / 4 |
| Diabetes | 0.755±0.043 | 0.736±0.021 | 15 / 13 | **0.766±0.024** | 0.746±0.039 | 15 / 11 |
| Adult Income | 0.839±0.006 | 0.841±0.005 | 17 / 15 | 0.842±0.004 | **0.842±0.005** | 30 / 13 |

Table 4: MP under 2-party VFL verses MP under 4-party VFL under 4 different settings of NN-based VFL using FedSGD communication protocol. "#Rounds" has the same meaning as in Tab. 3.

| Dataset | | aggVFL, 2-party | aggVFL, 4-party | splitVFL, 2-party | splitVFL, 4-party |
|---|---|---|---|---|---|
| CIFAR10 | MP | 0.790±0.003 | 0.747±0.003 | **0.798±0.010** | 0.762±0.003 |
| | #Rounds | 244±16 | 205±12 | 238±14 | 173±3 |
| CIFAR100 | MP | **0.454±0.006** | 0.417±0.008 | 0.423±0.005 | 0.382±0.004 |
| | #Rounds | 130±11 | 124±2 | 125±2 | 100±1 |

Table 5: MP and execution time under 2 different types of tree-based VFL.

| Dataset | | Random Forest w/o Encryption | XGBoost w/o Encryption | Random Forest w/ Encryption | XGBoost w/ Encryption (a.k.a. SecureBoost) |
|---|---|---|---|---|---|
| Credit | MP | 0.816±0.005 | 0.816±0.004 | 0.816±0.005 | 0.816±0.004 |
| | Exec.Time [s] | 138±4 | 366±16 | 410±10 | 881±6 |
| Nursery | MP | 0.884±0.010 | 0.890±0.011 | 0.884±0.010 | 0.890±0.011 |
| | Exec.Time [s] | 29±2 | 69±4 | 243±5 | 1194±21 |

**Model Partition.** The splitVFL setting yields a comparable or slightly higher MP compared to aggVFL on most datasets, due to the additional trainable layer serving as global model, evidenced by results from Tabs. 3, 4 and 7.

**Communication Protocols.** As shown in Tab. 3 and Tab. 6, compared to FedSGD, FedBCD and CELU-VFL exhibit comparable MP across all datasets with fewer communication rounds, supporting their efficacy in reducing communication overhead. Quantize and Top-k compress the transmitted data and successfully reduce the communication cost per round, but may result in an increase in communication rounds.

**Encryption.** For tree-based VFL, we consider two models with and without Paillier Encryption using 512-bit key size in Tab. 5. Note that XGBoost with Paillier Encryption is equivalent to SecureBoost (Cheng et al., 2021). Although MP values are consistent regardless of encryption, the execution time experiences a notable increase of 3 to 20 times when encryption is applied due to the additional encryption and decryption process.

**Number of Participants.** Impact of number of participants are shown in Tab. 4. A slightly lower MP is achieved using fewer communication rounds as the number of participants increases, demonstrating the increasing challenges brought by multi-party collaboration.

**Model Architectures.** Tabs. 5 and 13 (in Appendix I.1) compare the MP of different model architectures and show that different model architectures result in slightly different MP for each dataset, highlighting the importance of selecting best performing model for each dataset.

**Real world datasets.** Additional results on Criteo (Guo et al., 2017), Avazu (Qu et al., 2018), Cora (McCallum et al., 2000) and News20 (Lang, 1995) datasets using domain specific models (e.g. Wide&Deep Model (Cheng et al., 2016) for Criteo and Avazu, GNN for Cora) are provided in Tab. 7 , as they are considered for typical VFL applications, such as in recommendation problems.

## 6.2 ATTACK AND DEFENSE PERFORMANCE

We demonstrate attack and defense results of VFL on the AP-MP graph for each attack on MNIST, CIFAR10 and NUSWIDE datasets under aggVFL setting using FedSGD protocol in Figs. 3, 11 and 12. Each point in the figure represents a (MP, AP) pair with the size of markers representing

Table 6: MP, communication rounds (#Rounds), amount of information exchanged per round (Amount) under different communication protocols of NN-based VFL under aggVFL setting. 'Total' column is the total amount that equals to #Rounds×Amount.

| | MNIST | | | | NUSWIDE | | | |
|---|---|---|---|---|---|---|---|---|
| | MP | #Rounds | Amount (MB) | Total (MB) | MP | #Rounds | Amount (MB) | Total (MB) |
| FedSGD | **0.972±0.001** | 150 | 0.156 | 23.438 | **0.887±0.001** | 60 | 0.039 | 2.344 |
| FedBCD | 0.971±0.001 | 113 | 0.156 | 17.656 | 0.882±0.001 | 26 | 0.039 | 1.016 |
| Quantize | 0.959±0.006 | 161 | 0.117 | 18.867 | 0.881±0.002 | 94 | 0.029 | 2.754 |
| Top-k | 0.968±0.001 | 150 | 0.148 | 22.266 | 0.887±0.001 | 60 | 0.037 | 2.227 |
| CELU-VFL | 0.971±0.002 | 105 | 0.156 | 16.406 | 0.880±0.001 | 25 | 0.039 | 0.977 |

Table 7: Comparison of aggVFL and splitVFL on MP, #Rounds, Amount, total communication cost, Exec.Time for reaching specified MP with 4 real-world datasets of NN-based VFL with FedSGD communication protocol.

| Dataset | aggVFL | | | | | splitVFL | | | | |
|---|---|---|---|---|---|---|---|---|---|---|
| | MP | #Rounds | Amount (MB) | Total (MB) | Exec.Time [s] | MP | #Rounds | Amount (MB) | Total (MB) | Exec.Time [s] |
| Criteo | 0.715±0.053 | 2 | 0.125 | 0.250 | 0.190±0.132 | 0.744±0.001 | 3 | 0.125 | 0.375 | 0.234±0.126 |
| Avazu | 0.832±0.001 | 5 | 0.125 | 0.625 | 0.517±0.185 | 0.832±0.001 | 9 | 0.125 | 1.125 | 1.203±1.516 |
| Cora | 0.721±0.004 | 11 | 0.145 | 1.591 | 0.205±0.085 | 0.724±0.012 | 13 | 0.145 | 1.880 | 0.270±0.082 |
| News20-S5 | 0.882±0.014 | 57 | 0.005 | 0.278 | 0.430±0.076 | 0.893±0.013 | 61 | 0.005 | 0.298 | 0.613±0.269 |

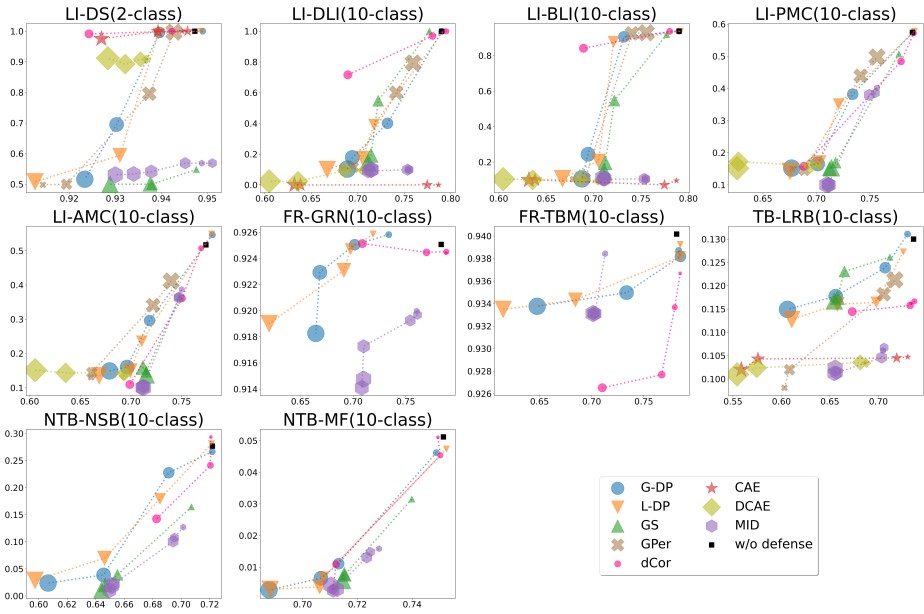

Figure 3: MPs and APs for different attacks under defenses [CIFAR10 dataset, aggVFL, FedSGD]

the relative magnitude of the corresponding defense hyper-parameter listed in Tab. 2. Note that although we try to provide comprehensive evaluation for various defenses, we do not force defense onto attacks, meaning that if a defense mechanism is designed for mitigating label inference attacks only, we do not assess its effectiveness against FR attacks or backdoor attacks.

We further rank all the defenses of different hyper-parameters based on their C-DCS. Due to space limitation, we show representative results for NUSWIDE dataset in Tab. 8. Full results are shown in Tabs. 15 to 17 in Appendix I.2 for MNIST, CIFAR10 and NUSWIDE dataset respectively, with calculation detail of each column provided in Appendix H.5.

**Attacks pose great threat to VFL.** Comparing the black squares illustrating the MP and AP of the attack against a VFL system without any defense in the sub-figures, we can observe that DS, DLI, BLI and TBM attacks are strong attacks with AP higher than 0.97, while MF attacks are quite weak

Table 8: T-DCS and C-DCS for All Defenses [NUSWIDE dataset, aggVFL, FedSGD]

| Defense Name | Defense Parameter | $T\text{-}DCS_{LI_2}$ | $T\text{-}DCS_{LI_5}$ | $T\text{-}DCS_{LI}$ | $T\text{-}DCS_{FR}$ | $T\text{-}DCS_{TB}$ | $T\text{-}DCS_{NTB}$ | $C\text{-}DCS$ |
|---|---|---|---|---|---|---|---|---|
| MID | 10000 | 0.7358 | 0.8559 | **0.8159** | 0.5833 | **0.7333** | 0.8707 | 0.7508 |
| MID | 1.0 | 0.7476 | 0.8472 | 0.8140 | 0.5833 | 0.7331 | 0.8700 | 0.7501 |
| MID | 100 | 0.7320 | 0.8536 | 0.8130 | 0.5833 | 0.7326 | **0.8711** | 0.7500 |
| G-DP | 0.1 | 0.7375 | 0.8262 | 0.7966 | 0.5863 | 0.7282 | 0.8675 | 0.7447 |
| L-DP | 0.1 | 0.7389 | 0.8177 | 0.7915 | 0.5863 | 0.7258 | 0.8603 | 0.7410 |
| MID | 0.1 | 0.7516 | 0.8259 | 0.8011 | 0.5833 | 0.7172 | 0.8563 | 0.7395 |
| MID | 0.01 | 0.7280 | 0.8092 | 0.7822 | 0.5844 | 0.7151 | 0.8627 | 0.7361 |
| dCor | 0.3 | **0.7641** | 0.8411 | 0.8155 | 0.5834 | 0.7289 | 0.8051 | 0.7332 |
| dCor | 0.0001 | 0.6496 | 0.6340 | 0.6392 | **0.5864** | 0.6307 | 0.8287 | 0.6712 |
| GS | 99.0 | 0.7404 | 0.8060 | 0.7841 | - | 0.6415 | 0.8408 | - |
| CAE | 1.0 | 0.6863 | 0.7822 | 0.7502 | - | 0.6830 | - | - |
| DCAE | 0.0 | 0.6669 | **0.8660** | 0.7996 | - | 0.6816 | - | - |
| GPer | 0.01 | 0.7386 | 0.8412 | 0.8070 | - | 0.7193 | - | - |

with AP below $0.1$. Other attacks, including NS (see Fig. 12 in Appendix I.2), PMC, AMC, GRN and LRB, also pose great threat to the VFL system.

**Defenses exhibit trade-offs between MP and AP.** For most of the attacks and defenses, we can observe an apparent trade-off between MP and AP, i.e. a lower AP is often gained with increasing harm of MP as defense strength grows, which can be controlled by adjusting defense hyper-parameters. An increase of noise level in DP-G and DP-L, sparsification rate in GS, regularization hyper-parameter $\alpha_d$ in dCor, confusional strength $\lambda_2$ in CAE and DCAE, regularization hyper-parameter $\lambda$ in MID or a decrease of DP budget $\epsilon$ in GPer will lead to lower MP and AP.

**DCS rankings are consistent across various datasets and settings.** As shown in Tabs. 15 to 17, the results of the C-DCS rankings are generally consistent across all $3$ datasets. In addition, the C-DCS ranking of the defense methods are still generally consistent even when the VFL model partition and communication protocol changes, as shown in Tabs. 15, 18 and 19 in Appendix I.2.As summarised in Fig. 22, these results demonstrate the robustness of the proposed DCS metrics, as well as the stableness of relative performance of different defense methods.Note that, T-DCS$_{FR}$ values are much lower than the T-DCS of other types, indicating that FR attacks are harder to defend than other attacks, which are consistent with human observation (see Fig. 13 in Appendix I.2.2).

**MID, L-DP and G-DP are effective on a wide spectrum of attacks.** MID demonstrates its capability of achieving a relatively lower AP while maintaining a higher MP compared to most other defenses as shown in Figs. 3, 11 and 12 and Tabs. 15 to 17; L-DP and G-DP are also generally effective under most attacks with above average T-DCS and C-DCS; DCAE is effective in defending against LI attacks; GS demonstrates strong defense ability for most of the LI attacks but performs less than satisfactory on LRB attacks; GPer performs similar to DP-G and DP-L in defending against label related attacks; dCor is less effective in limiting AP under NTB attacks but is largely effective against PMC and AMC attacks as shown in Figs. 3, 11 and 12.

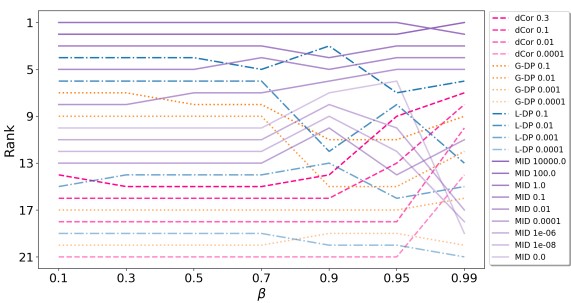

Figure 4: Change of C-DCS ranking with the change of $\beta$. [MNIST dataset, aggVFL, FedSGD]

**Change in $\beta$ does not significantly impact the C-DCS ranking.** $\beta$ in Eq. (1) represents users' trade-off preference on AP and MP when evaluating defenses, and can be adjusted. Here we use $\beta = 0.5$ for our main results. Figs. 4, 14 and 15 show the change of the ranking results with the change of $\beta$. Overall the relative rankings are not significantly impacted by $\beta$, demonstrating the stableness of the comparison results among various defenses. As $\beta$ grows to large values, e.g. $\geq 0.9$, the metric places overly strong weight on MP, resulting in more variations on the rankings. Specifically, dCor ranks higher with the increase of $\beta$ thanks to its better MP preservation at the cost of a weaker AP limitation.

**splitVFL is less vulnerable to attacks than aggVFL.** Using DCS metrics, we directly compare all the aforementioned attacks and defenses under aggVFL and splitVFL settings to understand the impact of changing the model partition strategy on VFL's vulnerability against attacks. We mainly use the DCS gap, defined as $DCS^{splitVFL} - DCS^{aggVFL}$ for each attack-defense point. Figs. 16 and 18

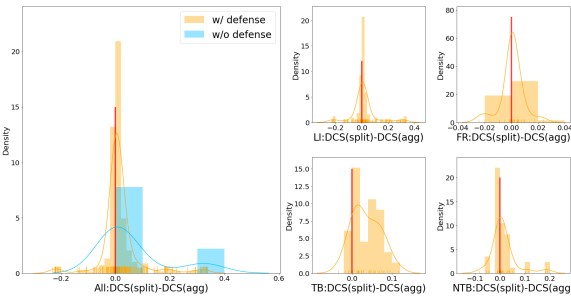

Figure 5: DCS gap Distribution, y-axis represents density [MNIST dataset, splitVFL/aggVFL, FedSGD]

(see Appendix I.2.4) show the DCS gap for all the attack-defense points among the 11 attacks and 8 defenses with all the evaluated parameters using MNIST and NUSWIDE dataset respectively. Figs. 5 and 17 displays the distribution of the DCS gaps depicted in Figs. 16 and 18 respectively. As all the black square points in Fig. 16 appear above or close to the red horizontal line at a value of 0.0 (see also the blue histograms that appear mostly at the right of the vertical line at a value of 0.0 in Fig. 5), we can conclude that splitVFL is less vulnerable to attacks than aggVFL when no defense is applied. In addition, splitVFL has an overall positive effect on boosting defense performance against attacks as well, as most of the DCS gap is positive in the last subplot of Fig. 16 when no attack is applied. Similar results can be seen from Figs. 17 and 18 in Appendix I.2.4. Further analysis are included in Appendix I.2.4.

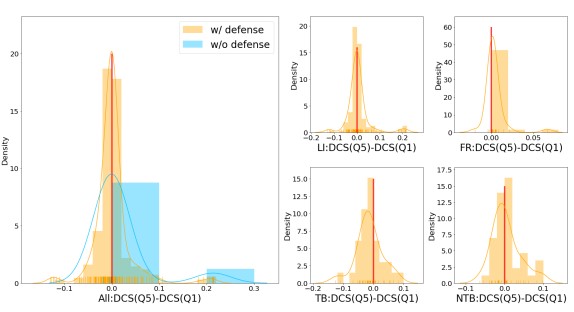

Figure 6: DCS gap Distribution, y-axis represents density [MNIST dataset, aggVFL, FedBCD/FedSGD]

**FedBCD is less vulnerable to attacks than FedSGD.** In addition, we compare DCS gap under FedSGD setting and FedBCD with $Q = 5$ to assess the impact of different communication protocols on model's vulnerability to attacks. DCS gap is defined as $\text{DCS}^{\text{FedBCD}} - \text{DCS}^{\text{FedSGD}}$ for each attack-defense point. Figs. 6 and 20 provides the distribution of the DCS gaps plotted in Figs. 19 and 21 (see Appendix I.2.5) which includes the DCS gap of all the attack-defense points among the 11 attacks and the 8 defenses using MNIST and NUSWIDE dataset respectively. As shown in Fig. 6, the blue histograms generally appear on the right of the vertical line of value 0.0, indicating that a system with FedBCD protocol is less vulnerable to attacks when no defense method is applied. In addition, a system with FedBCD also has an overall positive effect on boosting defense performance against FR and NTB attacks. This is evidenced by the fact that that the majority of DCS gaps are positive for FR and NTB attacks as shown in Fig. 6. Similar conclusions can be drawn from Figs. 20 and 21. Further analysis are included in Appendix I.2.5.

## 7 CONCLUSIONS AND LIMITATIONS

In this work, we introduce a light-weight VFL framework `VFLAIR` that implements basic VFL training and evaluation flow under multiple model partition, model architectures,communication protocols and attacks and defenses algorithms using datasets of different modality. We also introduce unified evaluation metrics and benchmark model utility, communication and computation efficiency, and defense performance under various VFL settings, which sheds lights on choosing partition, communication and defense techniques in practical deployment. Currently, the library has limited implementations on cryptographic techniques. Combination of non-cryptograhic and cryptographic techniques would be an interesting next step and we plan to add more advanced privacy-preserving and communication-efficient methods to our library.

## 8 REPRODUCIBILITY STATEMENT

We include all the hyper-parameters used for the benchmark experiments in Appendix H. Our code is also available at `https://github.com/FLAIR-THU/VFLAIR`).

## ACKNOWLEDGEMENT

This work was supported by the National Key R&D Program of China under Grant No.2022ZD0160504, the Tsinghua University(AIR)-Asiainfo Technologies (China) Inc. Joint Research Center, and Tsinghua-Toyota Joint Research Institute inter-disciplinary Program.

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

## A    RELATED WORK

**Vertical Federated Learning.** Federated Learning (FL) (McMahan et al., 2016b; Yang et al., 2019b;a) is a learning paradigm which allows multiple parties to build a machine learning model collaboratively without centralizing each data owner's local private data. Depending on whether data are partitioned by sample or by feature, FL can be further categorized into Horizontal FL (HFL) and Vertical FL (VFL) (Yang et al., 2019a). VFL (Liu et al., 2022b; Cheng et al., 2021; Jiang et al., 2022b) is commonly applied in real-world applications in the field of finance and advertising (Cheng et al., 2020; FedAI-maintainers) where cross-silo participants holding different sets of features of a common group of users jointly build machine learning models while keeping both local data and local model private. In VFL, only one participant possesses sensitive label information and is often referred to as *active party* while others are referred to as *passive parties*. Depending on how model is partitioned among parties, the VFL architecture can be further categorized into aggVFL and splitVFL (Liu et al., 2022a). In aggVFL, each party possesses a local sub-model, and a non-trainable aggregation function is used as the global model; while in splitVFL, a trainable aggregation model is used. Communication efficiency issue is a key bottleneck in VFL training. Approaches such as FedBCD (Liu et al., 2022b), Compressed-VFL (Castiglia et al., 2022b) (including Quantize and Top-k compression), Flex-VFL (Castiglia et al., 2022a), AdaVFL (Zhang et al., 2022), and CELU-VFL (Fu et al., 2022b) enhance VFL system efficiency by enabling each party to perform multiple local updates during each communication iteration. Although raw data and model parameters are not shared in NN-based VFL training and inference procedure, the threats of data leakage and model integrity remain. Existing attacks target either the reconstruction of private data (Li et al., 2022b; Jiang et al., 2022a) or the compromise of model robustness (Liu et al., 2020b; Zou et al., 2022; Pang et al., 2023). Private label or private features are both potential targets for data leakage attacks. Sample-level gradients (SLI) (Li et al., 2022b; Fu et al., 2022a; Sun et al., 2021a; Yang et al., 2022a) or batch-level gradients (BLI) (Zou et al., 2022) or trained local models (Fu et al., 2022a) can all be exploit to conduct label inference attacks. Model inversion technique (Jin et al., 2021; Luo et al., 2021; Li et al., 2022a; Jiang et al., 2022a) for white-box or black-box oracle setting with image or tabular data; or linear equation solving technique (Ye et al., 2022b;a) for black-box setting with binary value data can be applied to conduct feature reconstruction attacks. Assigning specific label to triggered samples (Zou et al., 2022) or adding noise to randomly selected samples (Liu et al., 2021b) or failing to transmit collaboration information (Liu et al., 2021b) can all result in successful backdoor attacks. These attacks pose significant challenges to VFL settings and necessitate effective countermeasures. To mitigate these threats, multiple defense methods have been proposed. Aside from cryptography techniques such as HE or Secure Multi-Party Computation (MPC) (Yang et al., 2019a), non-cryptography techniques such as reducing information by Adding Noise (Dwork, 2006; Li et al., 2022b; Fu et al., 2022a; Zou et al., 2022), Gradient Sparsification (Aji &Heafield, 2017), Gradient Discretization (Dryden et al., 2016; Fu et al., 2022a), Gradient Compression (Lin et al., 2018), or the combination of these information reduction techniques (Shokri &Shmatikov, 2015; Fu et al., 2022a) are often applied. Additionally, emerging defense mechanisms have been proposed either for specific attacks or multiple type of attacks. Disguising labels (Zou et al., 2022; Jin et al., 2021), label DP (Ghazi et al., 2021; Yang et al., 2022a), Dispersed Training (Wang et al., 2023) have proven effective in defending against label inference attacks; Dropout and Rounding (Luo et al., 2021), Adversarial Training (Sun et al., 2021b; Li et al., 2022a), Fabricated Features (Ye et al., 2022b) are proposed to prevent input feature leakage; Robust Feature Recovery (Liu et al., 2021b) has demonstrated its effectiveness for eliminating backdoor attacks, while Mutual Information Regularization (Zou et al., 2022) and Distance Correlation Regularization (Vepakomma et al., 2019; Sun et al., 2022) seek to mitigate various types of attacks.

**Tree-based VFL.** Due to its exceptional model performance and its inherent explainability, tree-based VFL has garnered widespread applications. Gradient Boosting Decision Tree (GBDT) is the most commonly employed approach for constructing trees-based VFL with various related algorithms having been proposed (Cheng et al., 2021; Chen et al., 2021; Fang et al., 2021; Tian et al., 2020; Xie et al., 2022). Another noteworthy tree-based ensemble algorithm in VFL is Random Forest (RF) (Ho, 1995) which leverages bagging and parallelism optimization techniques to improve training and inference efficiency (Liu et al., 2020a; Yao et al., 2022). While most tree-based VFL methods employ cryptographic approaches to protect sensitive data, some encrypt only partial information to enhance efficiency (Cheng et al., 2021; Wu et al., 2020) but results in data leakage threats (Chamani &Papadopoulos, 2020; Takahashi et al., 2023). Thus, various methods have been investigated aiming to solve the privacy (Cheng et al., 2021; Chen et al., 2021; Feng et al., 2019; Fang et al., 2021; Tian et al., 2020; Li et al., 2022e) and efficiency (Li et al., 2022e) issue.

## B  VFL FRAMEWORK

We include the training algorithm of NN-based VFL (Algorithm 1) as well as the description and algorithm (Algorithm 3) of the training and inference process of tree-based VFL in this section.

---

**Algorithm 1** A Basic VFL Training Procedure using FedSGD.

---

**Input**: learning rates $\eta_1$ and $\eta_2$
**Output**: Model parameters $\theta_1, \theta_2 \ldots \theta_K, \varphi$

1: Party $1,2,\ldots,K$, initialize $\theta_1, \theta_2, \ldots \theta_K, \varphi$.
2: **for** each iteration $j = 1, 2, \ldots$ **do**
3:     Randomly sample a mini-batch of samples $\{\mathbf{x}, \mathbf{y}\} \subset \mathcal{D}$ of size n;
4:     **for** each party $k$=1,2,\ldots,$K$ in parallel **do**
5:         Party $k$ computes $H_k = G_k(\mathbf{x}_k, \theta_k)$;
6:         Party $k$ sends $H_k$ to active party $K$;
7:     **end for**
8:     Active party computes the prediction $\hat{\mathbf{y}} = F(H_1, \ldots, H_K, \varphi)$ and loss $\mathcal{L} = \frac{1}{n}\ell(\mathbf{y}, \hat{\mathbf{y}})$;
9:     Active party $K$ updates $\varphi^{j+1} = \varphi^j - \eta_1 \frac{\partial \ell}{\partial \varphi}$;
10:    Active party $K$ computes and sends $\frac{\partial \mathcal{L}}{\partial H_k}$ to all other parties;
11:    **for** each party $k$=1,2,\ldots,$K$ in parallel **do**
12:        Party $k$ computes $\nabla_{\theta_k}\mathcal{L} = \frac{\partial \mathcal{L}}{\partial \theta_k} = \frac{\partial \mathcal{L}}{\partial H_k}\frac{\partial H_k}{\partial \theta_k}$;
13:        Party $k$ updates $\theta_k^{j+1} = \theta_k^j - \eta_2\nabla_{\theta_k}\mathcal{L}$;
14:    **end for**
15: **end for**

---

We also support using Homomorphic Encryption (HE) with Paillier Encryption to protect transmitted results during training VFL  Zou et al. (2022) (see Algorithm 2). In this protocol, Homomorphic Encryption (HE), denoted as $[[]]$, is applied to the communicated information, i.e., local model outputs $H_k$ in forward propagation and respective gradients $\frac{\partial \ell}{\partial H_k}$ in backward propagation. The active party then computes the gradient of loss with respect to $[[H_k]]$ under encryption, i.e. $[[\frac{\partial \mathcal{L}}{\partial H_k}]]$, and sends the results back to the passive parties for gradient updates. Note Taylor Expansion is used for gradient approximation as Paillier Encryption supports only addition and multiplication, following (Liu et al., 2018). The computed encrypted gradients $[[\frac{\partial \mathcal{L}}{\partial H_k}]]$ are subsequently added with a random local mask and transmitted to a Trusted Third Party (TTP) for decryption. The public key for the encryption is generated and distributed to each party by TTP, while the paired private keys are kept at TTP for decryption.

---

**Algorithm 2** A vertical federated learning framework with Homomorphic Encryption (Zou et al., 2022)

---

**Require:** Learning rate $\eta$
**Ensure:** Model parameters $\theta_1, \theta_2, \ldots, \theta_K$

1: Party $1, 2, \ldots, K$ initializes $\theta_1, \theta_2, \ldots, \theta_K$
2: Trusted Third Party (TTP) creates encryption pairs, sends public key to each party
3: **for** $j = 1, 2, \ldots$ **do**
4:     **for** each passive party $k \neq K$ in parallel **do**
5:        $k$ computes, encrypts, and sends $[[H_k]]$ to the active party.
6:     **end for**
7:     Active party $K$ computes and sends $[[\frac{\partial \mathcal{L}}{\partial H_k}]]$ to all other parties
8:     **for** each party $k = 1, 2, \ldots, K$ in parallel **do**
9:        $k$ computes $[[\frac{\partial \mathcal{L}}{\partial \theta_k}]]$ and sends them with a random mask to TTP for decryption.
10:       $k$ receives and unmasks $\nabla_{\theta_k}\mathcal{L} \leftarrow \frac{\partial \mathcal{L}}{\partial \theta_k}$ and updates $\theta_k^{j+1} = \theta_k^j - \eta\nabla_{\theta_k}\mathcal{L}$
11:     **end for**
12: **end for**

---

In tree-based VFL, the active party first broadcasts the set of record indices for the current node. Next, each passive party calculates the percentiles for each feature based on those indices. The passive party

then proceeds to create binary splits for each feature by comparing the feature value of each instance to the percentile values. After that, the passive party sends back the statistics of each split necessary for evaluation such as purity to the active party, and the active party selects the best split using a specific evaluation function. To be precise, Random Forest employs gini impurity for classification, while XGBoost utilizes its gain function, which is based on the gradient and hessian. Finally, the active party requests the owner of the best split to send the set of record indices for the children nodes generated by the best split. Tree-based VFL system continues these procedures recursively until certain stop conditions, like depth constraints, are satisfied. Algorithm 3 demonstrates training details of tree-based VFL.

---

**Algorithm 3** A Basic Training Process of Tree-based VFL.

---

**Input**: Evaluation function for a split
**Output** Trained trees:
 1: **for** each tree $j = 1, 2, ...$ **do**
 2:     Root Node $\leftarrow$ randomly sample a subset of record indices;
 3:     Nodes = [Root Node] {A list of nodes to be divided}
 4:     **while** Nodes is not empty **do**
 5:         Current node $\leftarrow$ pop one element of Nodes
 6:         **if** Current node satisfies the terminate conditions **then**
 7:             Continue
 8:         **end if**
 9:         Active party broadcasts the set of record indices of the current node to all other parties;
10:         **for** each party $k$=1,2,...,$K$ in parallel **do**
11:             Party $k$ receives the set of record indices of a node to be divided;
12:             Party $k$ calculates the statistics for all possible splits and sends them to the active party;
13:         **end for**
14:         Active party gathers the statistics of possible splits and picks the best one;
15:         **if** Party $k$ is selected as the best party **then**
16:             Party $k$ sends the sets of record indices of children nodes generated by the best split;
17:         **end if**
18:         Active party receives the sets of record indices of children nodes and appends them to Node;
19:     **end while**
20: **end for**

---

## C    QUICK GUIDE TO USE AND EXTEND VFLAIR

In this section, we give a step-by-step user guidance on how to use and extend `VFLAIR`.

**How to Use.**    Using `VFLAIR` requires the following steps:

1. **Build.** Download our code repository from GitHub and prepare all the required environments using commands shown in Fig. 7(a). Hardware requirements are listed in Tab. 9 in Appendix D.

2. **Prepare Dataset.** Prepare or download the dataset into folder `../../share_dataset/`, the default folder that contains all the datasets for experiments.

3. **Configure.** Modify the configuration JSON file under `./src/configs/` folder to specify settings, including learning hyper-parameters (e.g. 'epochs' and 'lr' for learning rate etc.), training dataset ('dataset'), training model ('model_list' for model of each party), model partition ('global_model' trainable or not), communication protocol ('communication') as well as attacks ('attack_list') and defense method ('defense'). Then rename (e.g. `my_config.json`) the configuration file and save it under `./src/configs/` folder. An example is shown in Fig. 7(b). Detailed explanations of all the parameters are provided in `./src/configs/README.md` for NN-based VFL and `./src/configs/README_TREE.md` for Tree-based VFL.

4. **Train.** Use command `cd src` and `python main_pipeline.py --configs my_config` consecutively to start a VFL training. Attack and defense evaluation will also be performed if specified in configuration file. MP, AP, communication rounds, amount of

```
# clone the repository
$ git clone <link-to-our-github-repo>

# install required packages
$ conda create -n VFLAIR python=3.8
$ conda activate VFLAIR
$ pip install --upgrade pip
$ cd VFLAIR
$ pip install -r requirements.txt

# install cuda related pytorch
$ pip install torch==1.10.1+cu113 torchvision==0.11.2+cu113 torchaudio==0.10.1 -f
  https://download.pytorch.org/whl/cu113/torch_stable.html
```

(a) Code download and environment preparation.

```
1   {
2       "epochs": 30,                                24      "communication":{
3       "lr": 0.01,                                  25          "communication_protocol": "Quantization",
4       "k": 2,                                      26          "iteration_per_aggregation": 1,
5       "batch_size": 2048,                          27          "quant_level": 16,
6       "dataset":{                                  28          "vecdim": 1
7           "dataset_name": "mnist",                 29      },
8           "num_classes": 10                        30      "defense": {
9       },                                           31          "name": "GaussianDP",
10      "model_list":{                               32          "parameters": {
11          "0": {                                   33              "party": [0],
12              "type": "MLP2",                      34              "dp_strength": 0.1
13              "input_dim": 392,                    35          }
14              "output_dim": 10                     36      },
15          },                                       37      "attack_list": {
16          "1": {                                   38          "0":{
17              "type": "MLP2",                      39              "name": "DirectLabelScoring",
18              "input_dim": 392,                    40              "parameters": {
19              "output_dim": 10                     41                  "party": [0]
20          },                                       42              }
21          "apply_trainable_layer": 0,              43          }
22          "global_model": "ClassificationModelHostHead"  44  }
23      },                                           45  }
```

(b) Example JSON file.

Figure 7: User guidance.

information transmitted each communication round as well as execution time for reaching the specified MP is recorded.

5. **Evaluate.** After training is finished, use `./src/metrics/data_process.ipynb` file to perform evaluations, e.g. calculate DCS, T-DCS, C-DCS.

We summarize the above steps in Fig. 8.

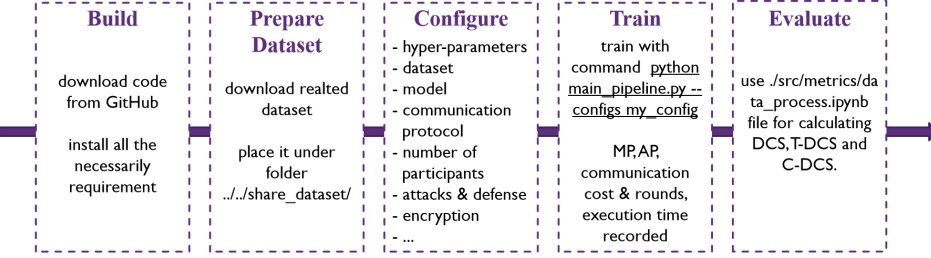

Figure 8: Step-by-step user guidance for using VFLAIR.

**How to Extend.** Extending our VFLAIR platform is also easy thanks to our modularized design of all the relative components.

- New hyper-parameter configurations (e.g. new training optimizer specification) can be added by modifying functions in `./src/load/LoadConfigs.py` file.

- New datasets and data partitioning methods can be added by first placing the raw data file under `../../share_dataset/` folder and then modifying functions in `./src/load/LoadDataset.py` file.

- New VFL models can be added under `./src/models/` folder and loaded via modifying the `./src/load/LoadModels.py` file with proper personalized modification in the 'model_list' part (see line 10-23 in Fig. 7(b)) of the configuration file.

- New communication protocols can be added by modifying the provided training flow in `./src/evaluates/MainTaskVFL.py` file to realize of the new communication protocol or by adding a new python file under the same directory of that file if the user finds the modification significant.

- New attacks can be added by inheriting `Attacker` class under `./src/evaluates/attacks/` folder for testing time attacks, or by modifying the main training file `./src/evaluates/MainTaskVFL.py` for training time attack.

- New defense methods can be added by modifying `./src/evaluates/defenses/defense_functions.py` file and the main training file `./src/evaluates/MainTaskVFL.py` if necessary.

- New party behaviors, e.g. new information exchange strategies for new communication protocols, can be added by modifying `./src/party/party.py`, under `./src/party/` folder.

**Use Case: Adding a new attack.** Below is a simple example for adding a new attack named 'BatchLabelReconstruction'.

1. New hyper-parameters for 'BatchLabelReconstruction' attack, e.g. attacker party ('party'), learning rate ('lr') and attack model training epochs ('epochs') in this case, can be added to the configuration file in the 'attack_list' part (line 29-37 in Fig. 9(a)). Rename the new configuration file, e.g. `my_new_config.json`, and save it under `./src/configs/`.

2. Implement necessary code for the new attack 'BatchLabelReconstruction' by inheriting `Attacker` class (see Fig. 9(b)) under `./src/evaluates/attacks/` folder for testing time attacks. Save it in file `./src/evaluates/attacks/BatchLabelReconstruction.py`. Note that, the name of the attacker class implemented in the file should be aligned with the name of file, 'BatchLabelReconstruction' in this case.

3. Run the new attack with command `python main_pipeline.py --configs my_new_config` under `./src/` folder.

**Use Case: Adding a new local model.** We further provide a second example here on adding a new local model 'MLP2_Softmax' to `VFLAIR`. Assume this local model is for the new attack 'BatchLabelReconstruction' added above.

1. New hyper-parameters for specifying the usage of 'MLP2_Softmax' model as local model for each party are 'type' attribute in 'model_list' part (line 12 and 17 in Fig. 9(a)), which should be specified as the name of the model. Then save the modified configuration file `my_new_config.json` still under `./src/configs/`.

2. New VFL bottom model 'MLP2_Softmax' can be implemented in `./src/models/mlp.py` like shown in Fig. 9(c). In this case, loading this new model does not need to modify `./src/load/LoadModels.py` file.

3. Run 'BatchLabelReconstruction' attack with new local model 'MLP2_Softmax' with command `python main_pipeline.py --configs my_new_config` under `./src/` folder.

## D  VFLAIR WORKFLOW

In this section, we present the key code modules and workflow of `VFLAIR`.

```json
1   {
2       "epochs": 30,
3       "lr": 0.01,
4       "k": 2,
5       "batch_size": 2048,
6       "dataset":{
7           "dataset_name": "mnist",
8           "num_classes": 10
9       },
10      "model_list":{
11          "0": {
12              "type": "MLP2_Softmax",  New Model
13              "input_dim": 392,
14              "output_dim": 10
15          },
16          "1": {
17              "type": "MLP2_Softmax",  New Model
18              "input_dim": 392,
19              "output_dim": 10
20          },
21          "apply_trainable_layer": 0,
22          "global_model": "ClassificationModelHostHead"
23      },
24      "communication":{
25          "communication_protocol": "FedSGD",
26          "iteration_per_aggregation": 1
27      },
28      "attack_list": {
29          "0":{
30              "name": "BatchLabelReconstruction",
31              "parameters": {
32                  "party": [0],
33                  "lr": 0.05,
34                  "epochs": 10000
35              }
36          }
37      }                                   New Attack
38  }
```

(a) New JSON file.

```python
class BatchLabelReconstruction(Attacker):
    def __init__(self, top_vfl, args):
        self.exp_res_path = ''

    def set_seed(self,seed=0):
        torch.backends.cudnn.benchmark = True

    def calc_label_recovery_rate(self, dummy_label, gt_label):
        return success / total

    def attack(self):
        return best_rec_rate
```

```python
class MLP2_Softmax(nn.Module):
    def __init__(self, input_dim, output_dim):
        super(MLP2_Softmax, self).__init__()
        self.layer1 = nn.Sequential(
            nn.Flatten(),
            nn.Linear(input_dim, 32, bias=True),
            nn.ReLU(inplace=True)
        )

        self.layer2 = nn.Sequential(
            nn.Linear(32, output_dim, bias=True),
            nn.Softmax(dim=1)
        )

    def forward(self, x):
        x = self.layer1(x)
        x = self.layer2(x)
        return x
```

(b) New attack.                    (c) New bottom model.

Figure 9: Adding new class.

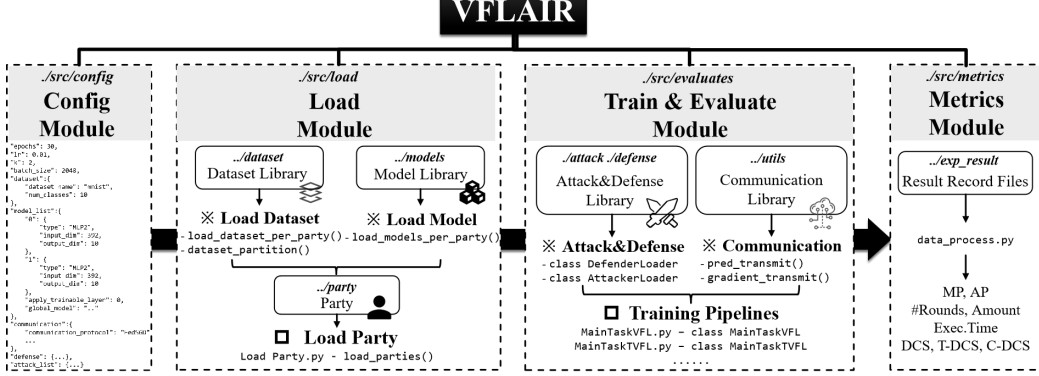

Figure 10: Code modules and Workflow of VFLAIR.

As shown in Fig. 10, VFLAIR contains 4 key modules, namely Config Module, Load Module, Train & Evaluate Module and Metrics Module.

The Config Module first processes the user-specified configurations (see Appendix C for detail), then passes it to the Load Module for party preparation. With function `load_parties()`, each party separately loads its dataset and local model with function `load_dataset_per_party()` and `load_models_per_party()` respectively.

Afterwards, the training pipeline starts in the Train & Evaluate Module. Communication protocols are implemented within the training pipeline and are controlled by `pred_transmit()` and `gradient_transmit()` functions. All the supported defense are integrated into the training pipeline. For attacks, training-time attacks (e.g. TB and NTB attacks) are integrated and evaluated within training pipeline while inference-time attacks that do not affect the training flow (e.g. LI and FR attacks) are launched and evaluated after the training pipeline. However, they are all prepared by an `AttackerLoader` object before the training pipeline starts.

Finally, the Metrics Module processes the recorded results using functions implemented in `data_process.py` file and produces evaluation metrics including MP, AP, communication rounds (#Rounds), amount of information transmitted between parties each round (Amount), execution time (EXec.Time), DCS, T-DCS and C-DCS.

We compared our framework `VFLAIR` with FATE, one of the most widely used FL platform supporting a broad range of VFL functionalities, on the system requirements for deployment in Tab. 9, to demonstrate that `VFLAIR` is a lightweight framework. According to FATE (https://github.com/FederatedAI/FATE/blob/master/deploy/standalone-deploy/README.md), deploying a stand-alone version of the FATE framework requires at least a 8 core CPU with 16G memory and 500G hard disk and the downloaded docker package for deployment is of size 4.92G for version 1.7.1.1. However, for our `VFLAIR`, a 1 core CPU with less than 4G memory and less than 4.0G hard disk is required for installation and environment preparation.

Table 9: Comparison of hardware requirements for installation between FATE and VFLAIR.

| | CPU | memory | Installation required hard disk |
|---|---|---|---|
| FATE (stand-alone, version 1.7.1.1) | 8 core | 16G | 4.92G |
| VFLAIR | 1 core | 4G | <4G |

## E   DETAIL DEFINITION OF ATTACK PERFORMANCE AND ATTACK PERFORMANCE FOR IDEAL DEFENSES

In this section, we give detailed definition of Attack Performance (AP) for each type of attack we studied in the literature. In this work, we use the AP for ideal defense AP*= 0.0 indicate a complete failure of the attack..

1. In **Label Inference (LI) Attacks**, AP is defined as *the ratio of correctly inferred labels* which can be regarded as the *label inference accuracy*. That is,

$$AP_{\text{LI}} = \frac{\text{Number of Samples Correctly Inferred}}{\text{Number of Total Samples}} \qquad (4)$$

   One exception is that for LI attacks under binary classification tasks, i.e. NS and DS (see Appendix F for explanation), we use *the AUC of inferred labels* instead for better measurement especially for unbalanced dataset. For both settings, high label inference accuracy indicates a successful attack.

2. In **Feature Reconstruction (FR) Attacks**, Mean Squared Error (MSE) is commonly employed as a measure to evaluate the quality of reconstruction, where a smaller MSE signifies a higher-quality reconstruction. So we define the AP for FR attacks as *a negative correlation of MSE with a constant shift*. Let $C$ denote the constant shift, $U_0$ be the real data stored in the attacked party, $U_{rec}$ be the data reconstructed by the attacker and function $\text{MSE}(\cdot, \cdot)$ for calculating the mean square error between the 2 function inputs, AP can be expressed as:

$$AP_{\text{FR}} = C - \text{MSE}(U_0, U_{rec}). \qquad (5)$$

   In our evaluation, we use $C = 1.0$ and normalize all raw input features to $[0.0, 1.0]$ before training and testing. In this case, $\text{MSE}(U_0, U_{rec}) = \mathbb{E}[(u_0^{(f)} - u_{rec}^{(f)})^2] \in [0.0, 1.0]$ where

$u_0^{(f)}, u_{rec}^{(f)}$ are the $f^{th}$ feature of $U_0, U_{rec}$ respectively. Thus, $AP_{\text{FR}} \in [0.0, 1.0]$ and a high AP indicates a strong attack.

3. In **Targeted Backdoor Attacks**, AP is defined as *the number of trigger inserted samples that is regarded as the target class by the VFL model to the total number of trigger inserted samples*, also *the backdoor accuracy*, achieved by the final VFL model, that is:

$$AP_{\text{TB}} = \frac{\text{Number of Triggered Samples Inferred as Target Class}}{\text{Number of Triggered Samples}} \quad (6)$$

Same as the 2 kinds of attacks above, a high AP indicates a successful backdoor attack.

4. In **Non-targeted Backdoor (NTB) Attacks**, *the gap of MP on attacked samples relative to the overall MP on all the samples* is defined as AP. That is:

$$AP_{\text{NTB}} = MP_{\text{all}} - MP_{\text{attacked\_sample}} \quad (7)$$

Still, a higher AP indicates a more successful attack.

# F   EVALUATED ATTACKS

In VFLAIR, all the attacks listed below are supported and evaluated in the benchmark experiments. Users can easily extend other attacks and evaluate them using our framework.

## F.1   LABEL INFERENCE (LI) ATTACKS

In **Label Inference (LI) Attacks**, the attacker (also the passive party) hopes to steal the sensitive label information kept at active part. There are multiple ways for the passive attacker to infer private labels. In this work, we evaluate 6 label inference attacks to assess the vulnerability of a VFL system under label leakage threat, including: Norm-based Scoring (NS) (Li et al., 2022b), Direction-based Scoring (DS) (Li et al., 2022b) and Direct Label Inference (DLI) (Li et al., 2022b; Zou et al., 2022) that exploit sample-level gradients, Batch-level Label Inference (BLI) (Zou et al., 2022) that uses batch-level gradients, Passive Model Completion (PMC) (Fu et al., 2022a) and Active Model Completion (AMC) (Fu et al., 2022a) that infer label from trained local models.

1. Norm-based Scoring (NS) (Li et al., 2022b). NS is designed for binary classification task, and is based on an experimental observation that the norm of gradient $||g||_2$ for a positive sample is generally larger than that of a negative sample in an unbalanced dataset. Thus the passive party can calculate the norm of sample-level gradients transmitted back by the active party and then infer the sensitive label information according to the norm values.

2. Direction-based Scoring (DS) (Li et al., 2022b). Similar to NS, DS also aims to infer label in binary classification task. DS is based on the observation that for any given sample pairs $x_a, x_b$, with their sample-level gradient denoted as $g_a, g_b$, the cosine similarity of these 2 gradients $cos(g_a, g_b) = g_a^T g_b / (||g_a||_2 ||g_b||_2)$ has a positive value if $x_a, x_b$ are of the same class, and a negative value if $x_a, x_b$ are of opposite classes. Attack can be launched using merely gradients that are transmitted back from the active party. However, to make the attacker more powerful, in the implementation of the attack, the gradient of one positive sample $g_+$ is additionally given.

3. Direct Label Inference (DLI) (Li et al., 2022b; Zou et al., 2022). DLI is based on the observation that at passive party, for sample $\{x_i, y_i\}$, $y_i^{th}$ element of the gradient transmitted back from active party $g_i$ is the only element having opposite sign compared to others, thus the label information is revealed.

4. Batch-level Label Inference (BLI) (Zou et al., 2022). There are cases where sample-level gradient information is protected, when using encryption techniques for example, and only batch-level gradient is revealed to passive party. BLI is designed to infer labels from this kind of setting. Inversion model is constructed and trained at passive party to invert label information from batch-level gradients.

5. Passive Model Completion (PMC) (Fu et al., 2022a). PMC exploits the information of label inherit in the trained local model at passive party. By adding a randomly initialized linear layer on top of the trained local model to get a "completion model" and fine-tune the "completion model" with auxiliary label data, the passive party is able to guess the label of each sample.

6. Active Model Completion (AMC) (Fu et al., 2022a). AMC is an enhanced version of PMC where the attacker (passive party) use a malicious local optimizer which adaptively scales up the gradient of each parameter in the adversary's bottom model. This results in a more accurate trained local model as the overall VFL model is tricked to rely more on the maliciously optimized local model at passive party. With a more accurate local model, passive party is expected to obtain a better attack using the same model completion techniques as PMC.

## F.2 Feature Reconstruction Attacks

In **Feature Reconstruction (FR) Attacks**, the attacker aims to recover other parties' local features from its local model and all information received from other innocent parties. Both active and passive party can be the attacker when labels are not needed for completing the attack. When neural network models serves as local models, targeted features are limited to binary values (Ye et al., 2022b;a) under a black-box setting in which the attacker neither has any knowledge of nor has access for querying the local model of the party under attack. Generation based on model inversion can be employed to reconstruct data by querying the trained model in a black-box oracle manner to recover tabular data (Luo et al., 2021). For white-box setting that has access to the trained model, model inversion techniques can also be used to recover image data (Jin et al., 2021) with the knowledge of trained local model, or with prior knowledge about data (Li et al., 2022a; Jiang et al., 2022a). In this work, we test the following 2 FR attacks:

1. Generative Regression Network (GRN) (Luo et al., 2021). By querying the trained VFL model, GRN attack reconstructs data features by matching the VFL prediction of real and reconstructed data. A generative model is trained to map random noise to targeted features.
2. Training-based Back Mapping by model inversion (TBM) (Li et al., 2022a). When auxiliary i.i.d. data of the local private data is available at the attacker, TBM utilizes these data to train a generative model in order to map the embedding feature of the victim party back to the original input feature. A strong assumption is used for TBM attack in which the attacker can query the whole trained VFL model with the data it obtained.

## F.3 Targeted Backdoor Attacks

**Targeted Backdoor (TB) Attacks** have a clear incorrect leading target and aims to manipulate the VFL model's behavior on samples marked with backdoor related features. We evaluated Label Replacement Backdoor (LRB) (Zou et al., 2022) in this work to assess the vulnerability of a VFL system under targeted backdoor threat.

1. Label Replacement Backdoor (LRB) (Zou et al., 2022) aims to assign an attacker-chosen label (target label) $\tau$ to input data with a specific pattern (i.e. a trigger). The passive attacker is assumed of knowing a few clean samples from the target class. The triggered poison samples are created locally by adding triggers to randomly selected samples from the passive attacker's own data. In training, the attacker replaces the embedding of a known clean sample from the target class with that of a triggered poison sample to replace the corresponding label of the poisoned sample.

## F.4 Non-targeted Backdoor Attack

Unlike TB attacks, **Non-targeted Backdoor (NTB) Attacks** only aim to affect the utility of the VFL model. Methods like adding noise to some randomly selected samples (Zou et al., 2022) or by adding missing features (Liu et al., 2021b) can be exploit during inference. The attacks used to assess the safety of a VFL system under NTB are listed below:

1. Noisy-Sample Backdoor (NSB) (Zou et al., 2022). In NSB attack, random noise $\sim \mathcal{N}(0, 2)$ is attached to arbitrarily selected samples by passive party to harm the model utility of VFL.
2. Missing Feature (MF) (Liu et al., 2021b). In MF attack, the embedding of passive party's local features of some samples are missing either due to the unstable network issue or due to the intentional hiding by the passive attacker. Missed features are treated as all $0$ in the implementation.

Detail attack hyper-parameters settings can be seen in Appendix H.3.

## G EVALUATED DEFENSES

In VFLAIR, all the defense methods listed in below are supported and evaluated in the benchmark experiments. Like for the attacks, users can easily extend other defense techniques and evaluate them using our framework.

1. Differential privacy with Gaussian noise (G-DP) (Dwork, 2006; Li et al., 2022b; Zou et al., 2022) or Laplace noise (L-DP) (Dwork, 2006; Fu et al., 2022a; Zou et al., 2022) added to gradients or local model predictions to defend against attacks launched at passive or active party.

2. Gradient Sparsification (GS) (Aji &Heafield, 2017; Fu et al., 2022a; Zou et al., 2022) defends against attacks by dropping elements in gradients that are close to $0$.

3. Gradient Perturb (GPer) (Yang et al., 2022a) defends against label leakage attacks by perturbing the gradients with the sum of gradients from each class added with random scalars. Label-DP is guaranteed using GPer.

4. Distance Correlation (dCor) (Sun et al., 2022; Vepakomma et al., 2019) defends against attacks by applying correlation regularization. When the passive party applies this defense to defend against feature reconstruction attacks, distance correlation is calculated between input features $X_k$ and local embedding $G(X_k, \theta_k)$ at each party in order to limit redundant information of $X_k$ kept in $G(X_k, \theta_k)$. In contrast, when active party applies this defense to defend against label inference attacks or backdoor attacks, distance correlation is calculated between label $Y$ and $G(X_k, \theta_k)$ of each party to limit redundant information of $Y$ kept in $G(X_k, \theta_k)$. $\log(dCor(X_k, G(X_k, \theta_k))$ and $\log(dCor(Y, G(X_k, \theta_k))$ are used in practice instead of $dCor(X_k, G(X_k, \theta_k)$ and $dCor(Y, G(X_k, \theta_k)$ to stabilize training.

5. Confusional AutoEncoder (CAE) (Zou et al., 2022) defends against label related attacks by disguising labels with an encoder and reconstruct the original label with the paired decoder. Confusion is added to map one class to multiple classes to further disguise label information.

6. Discrete-gradient-enhanced CAE (DCAE) (Zou et al., 2022) defends against attacks by applying discrete gradients along with CAE to get a stronger defense.

7. Mutual Information regularization Defense method (MID) (Zou et al., 2023) defends against attacks by limiting the information of label contained in local embedding.

Detailed hyper-parameters of these defenses can be found in Appendix H.4.

## H EXPERIMENTAL SETTINGS

We mainly use NVIDIA GeForce RTX 3090 for all the benchmark experiments except for tree-based VFL related experiments for which we use Intel(R) Xeon(R) CPU E5-2650 v2 instead. All the experiments are repeated $5$ times with different random seeds. The stopping criterion is determined as reaching a predefined number of epochs, while ensuring convergence at the same time. The reported results include both the mean values and the corresponding standard deviations. Other experimental settings are listed below.

### H.1 MODELS AND DATASETS

We construct our benchmark experiments on the following 9 datasets. The local models used for each dataset are listed in Tab. 10. Additionally, for the splitVFL setting, we employ a 1-layer MLP model as the global model for datasets other than Cora, for which a 1-layer graph convolution layer is adopted to pair with the GCN it adopts as local model, while a non-trainable global softmax function is used for the aggVFL setting.

- MNIST (Yann LeCun). The MNIST dataset comprises handwritten digits and consists of a training set with $60,000$ examples, along with a test set containing $10,000$ examples, distributed across 10 classes. All samples have been standardized in size and centered within $32 \times 32$ pixel grayscale images. In our experiments, we utilize the entire dataset, and each sample is horizontally divided into equal halves and assigned to respective parties.

Table 10: Summary of evaluated datasets under NN-based VFL. In "#Samples" column, the values denote the number of training and testing samples separately. In "Feature Partition" column, if the number of features is equal among each party, we present only one number; '[]' marks the number of features after extending discrete feature to one-hot features; '/' is used to separate the feature partition for 2-party VFL and 4-party VFL, otherwise the number stands for the feature partition of 2-party VFL. In "#Parameters / #Nodes" column, the value denotes the number of trainable parameters for each local model for neural network or logistic regression local model and denotes the number of nodes in total of each side for tree-based local model. $I$ and $C$ stands for the number of input features specified in Appendix H.1 and #Class respectively. (p) and (a) refer to passive and active party respectively.

| Dataset | #Class | #Samples | Feature Partition | Local Model (active & passive party) | #Parameters / #Nodes |
|---|---|---|---|---|---|
| MNIST (Yann LeCun) | 10 | 60000; 10000 | 392 | MLP-2 ($I$-32-$C$) | 12.9K |
| CIFAR10 (Krizhevsky &Hinton, 2009) | 10 | 50000; 10000 | 512 / 258 | Resnet18 | 11.17M |
| CIFAR100 (Krizhevsky &Hinton, 2009) | 100 | 50000; 10000 | 512 / 258 | Resnet18 | 11.17M |
| NUSWIDE (Chua et al., 2009) | 5 | 60000; 40000 | 1000 (p), 634 (a) | MLP-2 ($I$-32-$C$) | 32.2K (p), 20.5K (a) |
| Breast Cancer (Street et al., 1993) | 2 | 454; 114 | 15 | MLP-2 ($I$-32-$C$) | 2.3K |
| Diabetes (Kahn) | 2 | 614; 154 | 4 | Logistic Regression | 10 |
| Adult Income (Becker &Kohavi, 1996) | 2 | 34153; 14637 | 7 [15 (p), 93 (a)] | MLP-4 ($I$-64-128-64-$C$) | 17.7K (p), 22.7K (a) |
| Criteo (Guo et al., 2017) | 2 | 183362; 1650263 | 13 (p), 26 (a) | Wide&Deep (Cheng et al., 2016) | 1040.1M (p), 73.2K (a) |
| Avazu (Qu et al., 2018) | 2 | 727722; 80857 | 9 (p), 13 (a) | Wide&Deep (Cheng et al., 2016) | 144.1M (p), 208.1M (a) |
| Cora (McCallum et al., 2000) | 7 | 140; 1000 | 716 | 2-layer GCN (Kipf &Welling, 2017) | 23.2K |
| News20-S5 (Lang, 1995) | 5 | 4000; 1000 | 49658 | MLP-5 | 3.2M |
| Credit (Dua &Graff, 2017) | 2 | 24000; 6000 | 12 (p), 11 (a) | Logistic Regression | 26 (p), 24 (a) |
| | | | | MLP-4 ($I$-100-50-20-$C$) | 7.4K (p), 7.3K (a) |
| | | | | RandomForest / XGBoost | 580 / 620 |
| Nursery (Dua &Graff, 2017) | 5 | 10368; 2592 | 4 | Logistic Regression | 40 (p), 65 (a) |
| | | | | MLP-3 ($I$-200-100-$C$) | 22.2K (p), 23.2K (a) |
| | | | | RandomForest / XGBoost | 402 / 415 |

- CIFAR10 (Krizhevsky &Hinton, 2009). The CIFAR10 dataset consists of $60,000$ colour images, each of size $32 \times 32$, of 10 classes, with $6,000$ images per class: $5,000$ for training and $1,000$ for testing. All the data are used in our experiments. Each sample is horizontally split in to equal halves and assigned to respective parties under 2-party VFL setting, or is equally split into 4 parts of size $16 \times 16$ and assigned to respective parties under 4-party VFL setting.

- CIFAR100 (Krizhevsky &Hinton, 2009). Similar to CIFAR10 dataset, CIFAR100 dataset consists of $60,000$ $32 \times 32$ colour images, but are distributed across 100 classes containing 600 images each: 500 for training and 100 for testing. All the data are used in our experiments. Each sample is horizontally split in to equal halves and assigned to each party under 2-party VFL setting, or is equally split into 4 parts of size $16 \times 16$ and assigned to each party under 4-party VFL setting.

- NUSWIDE (Chua et al., 2009). NUSWIDE dataset is a web image dataset that includes: (1) $269,648$ images and the associated tags from Flickr; (2) 5 types of low-level features extracted from the images and 1 bag of words feature of 500 dimension; (3) ground-truth label for 81 concepts. In our experiments, we use all the 5 low-level image features, that is a total of 634 features, for the active party and use the top 1000 frequent tags associated to the images for the passive party. We use only data from 'buildings', 'grass', 'animal', 'water' and 'person' in our experiments. For binary classification tasks, we use only data from 'clouds' and 'person'.

- Breast Cancer (Street et al., 1993). Breast Cancer dataset is a tabular dataset consists of 30 statistical descriptions of 568 breast tumors which are categorized as either malignant (cancerous) or benign(non-cancerous). In our experiments, we use $20\%$ of the whole dataset samples for testing and the rest for training. Each party possesses 15 statistical description features.

- Diabetes (Kahn). The Diabetes dataset comprises 768 samples, each accompanied by 8 diagnostic measurements corresponding to individual patients, along with the diagnosis indicating whether the patient has diabetes. We also use $20\%$ of the whole dataset samples for testing and the rest for training. Each party possesses 4 diagnostic measurements.

- Adult Income (Becker &Kohavi, 1996). The Adult Income dataset consists of 14 distinct features, including demographic information such as age, education, and occupation, collected from a dataset of $48,790$ individuals. The primary task associated with this dataset is to predict whether a person earns an annual income exceeding 50K. We use $30\%$ of the whole

dataset samples for testing and the rest for training. Category features are first changed to one-hot features before sending into the model for prediction. Passive party obtains the 6 non-category features as well as a category feature 'workclass', while the active party controls the rest 7 category features. After extending the categorical discrete features, 15 and 93 features are kept separately at each party.

- Criteo (Guo et al., 2017) The Criteo dataset contains real world click-through data of display advertisements served by Criteo of 7 days and whether the advertisement has been clicked or not. The primary task is to predict whether clicktion is done. Only $1,833,625$ samples from the whole dataset is used in our experiments with $90\%$ used for training and the rest $10\%$ used for testing following previous work (Fu et al., 2022c). Each sample has 26 anonymous categorical features assigned to passive party and 13 continuous features assigned to active party also following previous work (Fu et al., 2022c).

- Avazu (Qu et al., 2018) The Avazu dataset contains 11 days real world click-through data from Avazu. The primary task is also to predict whether clicktion is done. We use only $808,579$ samples from the whole dataset in our experiments with $90\%$ for training and rest $10\%$ for testing following previous work (Fu et al., 2022c). Each sample has 9 anonymous categorical features assigned to passive party and 13 categorical features that are not anonymous assigned to active party also following previous work (Fu et al., 2022c).

- Cora (McCallum et al., 2000) The Cora dataset is a citation network dataset with nodes representing computer science research papers and edges representing citations between them. The task is to predict the category of a node, which falls into one of the following 7 categories: 'Neural_Networks', 'Probabilistic_Methods', 'Genetic_Algorithms', 'Theory', 'Case_Based', 'Reinforcement_Learning' and 'Rule_Learning'. A total of 2708 nodes are provided in the dataset, with 140 nodes with labels and are used as training data. We use 1000 of the rest for testing. A total of 1432 features are provided for each node and we split them equally and signed to each party.

- News20-S5 (Lang, 1995) The News20 dataset, also the 20 Newsgroups dataset, is a collection of approximately $20,000$ newsgroup documents, partitioned (nearly) evenly across 20 different newsgroups. We use only the first 5 categories, namely 'alt.atheism', 'comp.graphics', 'comp.os.ms-windows.misc', 'comp.sys.ibm.pc.hardware' and 'comp.sys.mac.hardware' for our experiments, resulting in a total of $5,000$ samples. We use $80\%$ of the data for training and the rest $20\%$ for testing. Each party controls $49,658$ features in our setting.

- Credit (Dua &Graff, 2017). Credit dataset comprises $30,000$ samples, each with 23 features, including attributes like age and education level. The primary objective here is to predict the likelihood of default payment. For our experimentation, we allocate $20\%$ of the entire dataset for testing, while the remaining $80\%$ is designated for training. While the passive party owns the 12 features related to the amount of bill statement and previous statement, the active party possesses the other 11 features related background information and repay statement.

- Nursery (Dua &Graff, 2017). Nursery dataset consists of $12,960$ samples of 8 features with the task of predicting the recommendation level of applications for nursery schools. We also use $20\%$ of the whole dataset samples for testing and the rest for training. While the active party owns 4 features related to family structure and financial standings, the passive party has other 4 features.

## H.2    VFL MAIN TASK TRAINING HYPER-PARAMETERS

We introduce the hyper-parameters uesd for MP evaluations and benchmarks showed in Tabs. 3 to 7 and 13 in the following.

For NN-based VFL, the learning rate and training epochs use for reporting the MP listed in Tabs. 3, 4, 6, 7 and 13 are included in Tabs. 11 and 12. A batchsize of 1024 is used throughout all the experiments (except for MNIST, Criteo, Avazau and News20-S5 which uses a batchsize of 2048, 8192, 8192, 128 respectively) and is not listed in the table. For tree-based VFL, for reporting the MP listed in Tab. 5, each party is equipped with a number of 5 trees each of depth 6 under all circumstances. Note that learning rate is only utilized for XGBoost and is set to 0.003 in the experiments.

Table 11: Hyper-parameters for Tabs. 3, 4, 7 and 13. LR is short for learning rate.

| Dataset | aggVFL, FedSGD | | aggVFL, FedBCD | | splitVFL, FedSGD | | splitVFL, FedBCD | |
|---|---|---|---|---|---|---|---|---|
| | LR | epochs | LR | epochs | LR | epochs | LR | epochs |
| MNIST | 0.01 | 30 | 0.005 | 30 | 0.01 | 30 | 0.005 | 30 |
| CIFAR10 (2-party) | 0.001 | 30 | - | - | 0.001 | 30 | - | - |
| CIFAR10 (4-party) | 0.001 | 30 | - | - | 0.001 | 1024 | - | - |
| CIFAR100 (2-party) | 0.01 | 30 | - | - | 0.01 | 30 | - | - |
| CIFAR100 (4-party) | 0.001 | 40 | - | - | 0.001 | 40 | - | - |
| NUSWIDE | 0.003 | 10 | 0.003 | 10 | 0.006 | 10 | 0.003 | 10 |
| Breast Cancer | 0.05 | 50 | 0.005 | 50 | 0.05 | 50 | 0.005 | 50 |
| Diabetes | 0.05 | 80 | 0.01 | 80 | 0.05 | 80 | 0.01 | 80 |
| Adult Income | 0.01 | 50 | 0.001 | 50 | 0.01 | 50 | 0.001 | 50 |
| Criteo | 0.0001 | 5 | - | - | 0.0001 | 5 | - | - |
| Avazu | 0.0001 | 10 | - | - | 0.0001 | 10 | - | - |
| Cora | 0.01 | 20 | - | - | 0.01 | 20 | - | |
| News20-S5 | 0.001 | 80 | - | - | 0.002 | 80 | | - |
| Credit (LR) | 0.6 | 300 | - | - | 0.6 | 300 | - | - |
| Credit (NN) | 0.01 | 40 | - | - | 0.01 | 40 | - | - |
| Nursery (LR) | 0.5 | 40 | - | - | 0.5 | 40 | - | - |
| Nursery (NN) | 0.01 | 40 | - | - | 0.01 | 40 | - | - |

Table 12: Hyper-parameters for Tab. 6. Note that the hyper-parameters for the FedSGD and FedBCD in Tab. 6 are the same as that in Tab. 3.

| Dataset | Quantize | | Top-k | | CELU-VFL | | |
|---|---|---|---|---|---|---|---|
| | LR | epochs | LR | epochs | LR | epochs | $\xi$ |
| MNIST | 0.03 | 30 | 0.05 | 30 | 0.008 | 30 | 0.5 |
| NUSWIDE | 0.003 | 10 | 0.003 | 10 | 0.008 | 10 | 0.8 |

## H.3 ATTACK HYPER-PARAMETERS

Our evaluated attacks can be categorized into **Label Inference** (LI) attacks, **Feature Reconstruction** (FR) attacks, **Targeted Backdoor** (TB) attacks and **Non-targeted Backdoor** (NTB) attacks. Each attack is launched separately on the VFL systems trained with the hyper-parameters listed above in Appendix H.2. Specific hyper-parameters for each attack is listed in below if exist.

- For LI attacks, attacker is the passive party. In **NS** (Li et al., 2022b), **DS** (Li et al., 2022b) and **DLI** (Li et al., 2022b; Zou et al., 2022), no other attack related hyper-parameter needs to be specified. However, for MNIST dataset, in order to achieve a higher MP, we use a learning rate of $0.001$ for VFL model training under binary classification tasks (DS attack). In **BLI** (Zou et al., 2022), learning rate and number of training epoch for the inference model are set to $0.05$ and $10000$ respectively. In **AMC** and **PMC** (Fu et al., 2022a), we randomly take $4$ training samples from each class to form the auxiliary labeled dataset for training of the "completion model". Both the trained local model and the classification head used to complete the model are fine-tuned using the auxiliary labeled dataset and some non-labeled data from the training dataset. Number of "completion model" training epochs, learning rate and training batchsize are set to $25, 0.002, 16$ separately for MNIST and CIFAR10 datasets and are set to $20, 0.002, 16$ separately for NUSWIDE dataset. Note that for better attack stability, we change the VFL main task learning epoch to $100$ for PMC and AMC under all the 3 datasets.

- For FR attacks, attacker is the active party. In **GRN** (Luo et al., 2021), following the original paper, the reconstruction model is trained for $60$ epochs with a batchsize of $1024$ for all datasets, as well as a learning rate of $0.005$ for MNIST dataset and $0.0001$ for CIFAR10 dataset. While in **TBM** (Li et al., 2022a), the reconstruction model is trained for $50$ epochs with a learning rate of $0.0001$ and a batchsize of $32$. Note that, since TBM requires auxiliary data with the same distribution as the training data, we use $10\%$ of the training data to form the auxiliary dataset and train the VFL model using only the rest $90\%$ following the original work (Li et al., 2022a).

- For TB attacks, **LRB** (Zou et al., 2022) is evaluated with the passive party being the attacker. Target class is randomly selected for each training. Following previous work (Zou et al., 2022), $1\%$ of the data is randomly selected and attached with specific trigger of 4 pixels to form backdoor samples for MNIST and CIFAR10 datasets, while samples that have value 1 at the last bit of the tag data, which take up less than $1\%$ of the whole dataset, are treated as backdoor samples for NUSWIDE dataset.

- For NTB attacks, attack is done by the passive party at inference time. In **NSB** (Zou et al., 2023), $1\%$ of the testing samples are randomly selected and added with random noise sampled from $\mathcal{N}(0, 2)$ by the attacker party. In **MF** (Liu et al., 2021b), a missing rate of $0.25$ is evaluated which means during inference, a quarter of the intermediate local model output from the passive attacker is replaced with **0** before transmitted to the active party for aggregation.

### H.4 DEFENSE HYPER-PARAMETERS

To ensure a comprehensive benchmark, we evaluate a total of 8 defense methods using our proposed benchmark pipeline. For each defense method, we systematically test different parameters to ensure a thorough and adequate evaluation. When defenses are applied to a VFL under a particular attack, all the hyper-parameters are not changed compared to that when that attack is applied. Detailed defense related experimental hyper-parameter settings are listed below:

- In **L-DP** or **G-DP** (Dwork, 2006; Li et al., 2022b; Zou et al., 2022), gradients are first 2-norm clipped with $0.2$ when the defense is applied at the active party. When passive party applies this defense, noises are added to normalized local prediction results. Then noise with "diversity" scale parameter or standard deviation ranging from $0.0001$ to $1.0$ is added to clipped gradients.

- In **GS** (Aji &Heafield, 2017; Fu et al., 2022a; Zou et al., 2022), drop rate ranging from $95.0\%$ to $99.5\%$ are evaluated in the experiments.

- In **GPer** (Yang et al., 2022a), we conduct tests with various standard deviations for the noise variable $u$ with the value of range $0.0001$ to $0.1$.

- In **dCor** (Sun et al., 2022; Vepakomma et al., 2019), we set distance correlation regularizer coefficient $\lambda$ from $0.0001$ to $0.3$ and record the corresponding results.

- In **CAE** (Zou et al., 2022), we consider various confusional levels with $\lambda_2$ value ranging from $0.0$ to $1.0$ while in **DCAE** (Zou et al., 2023), we select the same set of confusional levels and fix the bin number for gradient discrete to 12.

- In **MID** (Zou et al., 2023), we test the scenario where both active party and passive party apply MID, which suit the real world application since both sides need to protect themselves against potential attacks. Hyper-parameter $\lambda$ that signifies the strength of the defense ranges from $0.0$ to $1000.0$.

### H.5 C-DCS CALCULATION DETAIL

T-DCS values and C-DCS values are all calculated with respect to 1 single dataset. All T-DCS scores, i.e. $T\text{-}DCS_{LI_2}, T\text{-}DCS_{LI_{10}}, T\text{-}DCS_{LI_5}, T\text{-}DCS_{LI}, T\text{-}DCS_{FR}, T\text{-}DCS_{TB}, T\text{-}DCS_{NTB}$ are calculated by averaging the DCS values of each evaluated attack belonging to that type under the particular dataset. Then C-DCS values are calculated with Eq. (3) using only $T\text{-}DCS_{LI}, T\text{-}DCS_{FR}, T\text{-}DCS_{TB}, T\text{-}DCS_{NTB}$.

## I ADDITIONAL EXPERIMENTAL RESULTS

### I.1 ADDITIONAL MAIN TASK PERFORMANCE RESULTS

We place the comparison results of MP and communication rounds (#Rounds) of NN-based VFL with different local models in Tab. 13 here in the appendix. Together with Tab. 5, we compare the performance of linear regression, tree and neural network model architecture under 2 different datasets.

Table 13: Comparison of MP and communication rounds using different local models in NN-based VFL with FedSGD communication protocol. We mainly follow (Ye et al., 2022b) for the selection of MLP model architecture.

| | | Credit | | Nursery | |
|---|---|---|---|---|---|
| | | Linear Regression | Neural Network (MLP-4) | Linear Regression | Neural Network (MLP-3) |
| aggVFL | MP | 0.820±0.001 | 0.826±0.001 | 0.938±0.001 | 0.999±0.001 |
| | #Rounds | 22±10 | 56±20 | 58±19 | 77±13 |
| splitVFL | MP | 0.821±0.000 | 0.826±0.001 | 0.931±0.005 | 0.999±0.001 |
| | #Rounds | 21±6 | 85±20 | 52±4 | 102±19 |

Table 14 presents the MP and running times with and without encryption. The number of epochs for Credit with LR is reduced to 40 since the training with HE takes a lot of time. The MP is different from those of Table 13 since the experiment of Table 14 uses the logit-loss even for multi-class classification. In terms of MP, a slight performance degradation occurs due to the approximation error caused by Taylor Expansion. The increase in running time due to encryption is more than 1000 times, as we do not currently support GPU for matrix operations on encrypted values.

Table 14: MP and execution time under aggVFL with Homomorphic Encryption.

| Dataset | | Linear Regression (w/o encryption) | Linear Regression (w/ encryption) |
|---|---|---|---|
| Credit | MP | $0.821 \pm 0.000$ | $0.805 \pm 0.002$ |
| | Exec.Time [s] | $3 \pm 0$ | $11149 \pm 357$ |
| Nursery | MP | $0.918 \pm 0.001$ | $0.912 \pm 0.001$ |
| | Exec.Time [s] | $1 \pm 0$ | $2971 \pm 57$ |

## I.2    Additional Attack and Defense Performance with More Datasets

Due to space limit, we place the plot for MPs and APs under MNIST (Fig. 11) and NUSWIDE (Fig. 12) datasets in this section with detailed analysis of the results. A brief summary of the below analysis is already included in Sec. 6.2.

### I.2.1    MP and AP

**MID** defense is capable of achieving a relatively lower AP while maintaining a similar MP compared to most other defenses, demonstrating its effectiveness in defending against a wide spectrum of attacks by reducing the information of label $Y$ and local feature $X_p$ kept in $H_p$ with a mutual information (MI) regularizer.

**dCor** targets at attacks that explore the information contained in $H_p$ for deducing $Y$ or $X_p$ by limiting the distance correlation between $H_p$ and $Y$ or $X_p$ for defending against attacks launched by passive or active party respectively. This defense ideology is similar to that of MID, which directly regularized the mutual information between $Y$ and $H_p$. MID performs better than dCor on gradient-based LI attacks like DS, DLI and BLI, likely due to the fact that limiting the MI between $H_p$ and $Y$ simultaneously limits the dependency between the correlated gradient $g_p$ at passive party and $Y$, which is better than reducing the distance correlation between $H_p$ and $Y$. Additionally, when compared to other attacks, dCor appears less effective in limiting AP under NTB attacks across all the 3 datasets which is reasonable since this defense is not designed for defending against attacks that introduce information loss into transmitted data.

**CAE** and **DCAE** focus on disguising label by mapping real one-hot labels to soft-fake labels. CAE consistently performs well across all datasets when defending against DLI, BLI and LRB attacks, which utilize merely the information of the current sample without any auxiliary information or data. For PMC and AMC attacks that rely on auxiliary label information, DCAE is more effective with the help of information reduction from quantization of gradients. The limited effectiveness of DCAE against DS attack can be attributed to the fact that the quantization is not effective in perturbing the direction of the gradient, while DS attack relies on the cosine similarity between the gradient of each sample and a known sample. These 2 attacks are not designed for FR and NTB attacks.

**GS** is a defense that injects information loss by sparsifying gradients, i.e. setting gradient elements with small absolute value to $0.0$, during training. As it is the gradients, which is directly related to labels but not input features, that GS modifies to defend against potential attacks, we do not benchmark its performance on FR attacks following previous works (Luo et al., 2021; Li et al.,

2022a). GS shows a strong defense ability for most of the LI attacks but exhibits less than satisfactory defense results against LRB attack, which replace the gradients of selected samples with those related to target labels to alter their labels. This is likely due to the fact that GS still preserves the gradient elements with larger absolute values which are responsible for learning the triggered patterns injected by the backdoor attack.

**_DP-G_ and _DP-L_** are defenses that inject noise to gradients or local model output to defend against attacks launched by passive or active party respectively. They show similar trend and display a clear trade off between MP and AP in all the attacks.

**_GPer_** targets at defending label inference attacks so we apply it only on label related attacks, i.e. LI and LRB attacks. GPer guarantees label-DP by perturbing the gradients and performs similarly to DP-G and DP-L in defending against these attacks, consistent with the original work (Yang et al., 2022a) which shows that GPer performs the same as adding isotropical noise to gradients.

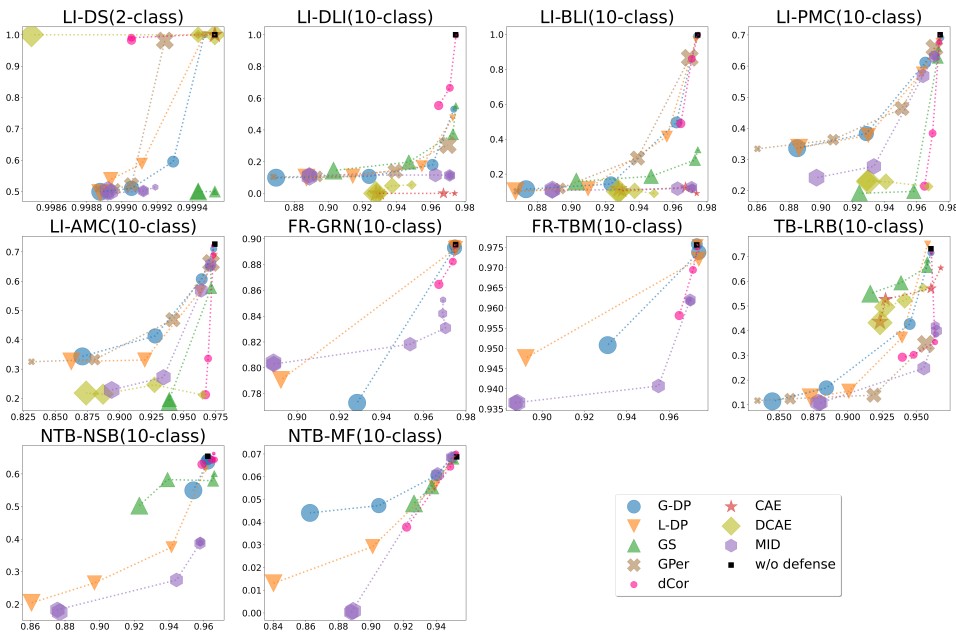

Figure 11: MPs and APs for different attacks under defenses [MNIST dataset, aggVFL, FedSGD]

### I.2.2 VISUALIZATION OF TBM ATTACK

We show the visualization of reconstructed features of TBM attack under various defenses with MNIST dataset in Fig. 13. All the defense hyper-parameters selected are the most effective one with highest DCS among the particular defense method, demonstrated in Fig. 11 (sub-figure for TBM) and Tab. 15. It's clear from Fig. 13 that, although the reconstructed images are noisy when defense exits, the contour can still be seen easily. This implies the difficulty for defending against TBM attack which exploits auxiliary i.i.d. data for feature reconstruction guidance.

### I.2.3 DCS, T-DCS AND C-DCS RANKING

We also place the full T-DCS and C-DCS table for MNIST (Tab. 15), CIFAR10 (Tab. 16), NUSWIDE (Tab. 17) datasets here due to limit of space. Basic analysis has already been included in Sec. 6.2. The overall relative rankings of all the defenses are still similar to that of other datasets.

We also show the change in C-DCS ranking under different $\beta$ for CIFAR10 and NUSWIDE here due to space limit. Similar results can be seen from Figs. 14 and 15 while the ranking fluctuates more

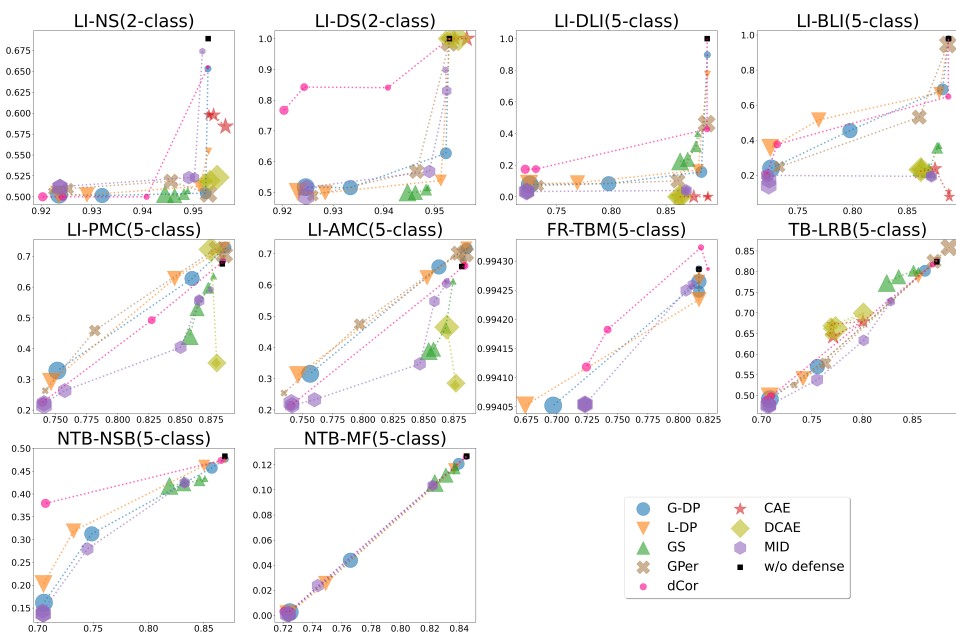

Figure 12: MPs and APs for different attacks under defenses [NUSWIDE dataset, aggVFL, FedSGD]

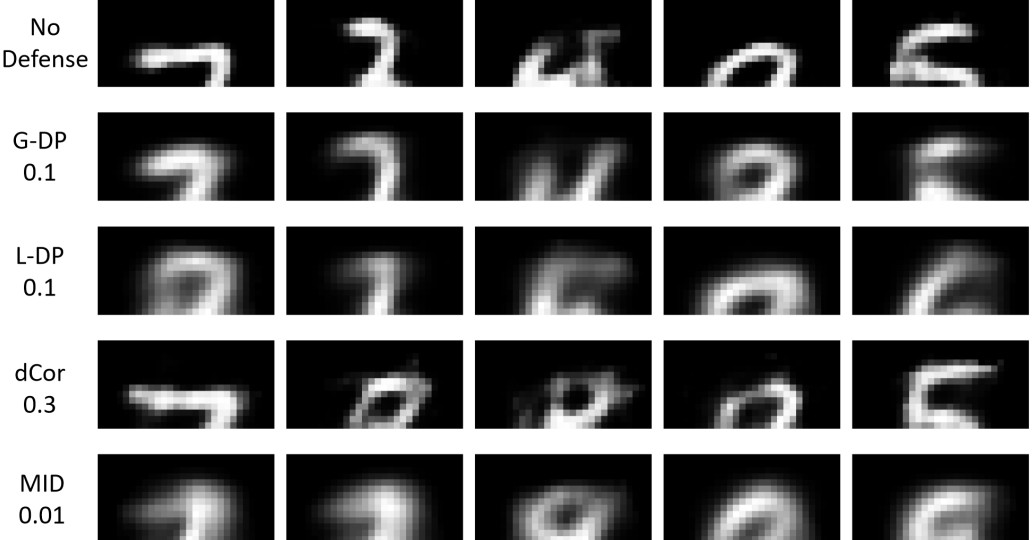

Figure 13: Visualization of the feature reconstruction results of TBM attack for MNIST dataset. All the defense hyper-parameters selected are the most effective ones with highest DCS among the particular defense method.

under NUSWIDE dataset compared to the other 2 datasets, probably because of the larger drop of MP for all the defenses when applied to NUSWIDE dataset.

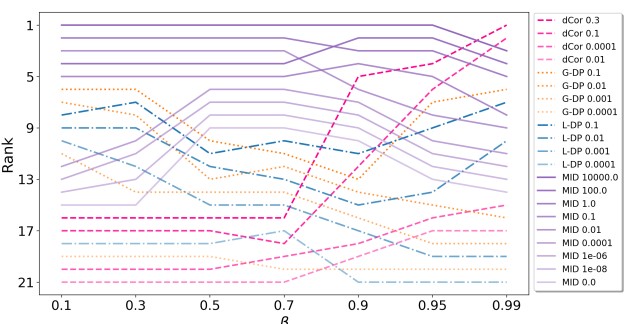

Figure 14: Change of C-DCS ranking with the change of $\beta$. [CIFAR10 dataset, aggVFL, FedSGD]

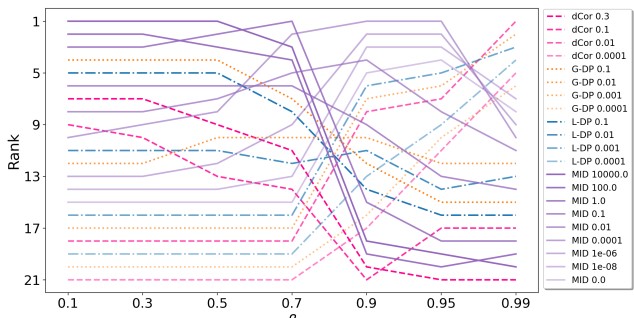

Figure 15: Change of C-DCS ranking with the change of $\beta$. [NUSWIDE dataset, aggVFL, FedSGD]

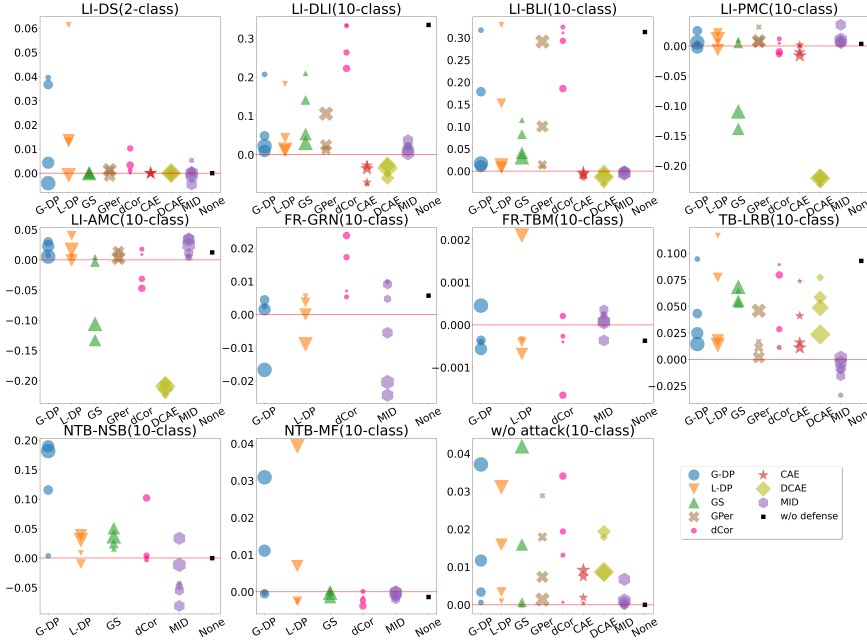

Figure 16: DCS gap for each attack-defense point [MNIST dataset, splitVFL/aggVFL, FedSGD]

Table 15: T-DCS and C-DCS for All Defenses [MNIST dataset, aggVFL, FedSGD]

| Defense Name | Defense Parameter | $T\text{-}DCS_{LI_2}$ | $T\text{-}DCS_{LI_{10}}$ | $T\text{-}DCS_{LI}$ | $T\text{-}DCS_{FR}$ | $T\text{-}DCS_{TB}$ | $T\text{-}DCS_{NTB}$ | $C\text{-}DCS$ | Ranking |
|---|---|---|---|---|---|---|---|---|---|
| MID | 100 | 0.7371 | 0.8810 | 0.8523 | 0.6184 | 0.9070 | **0.9093** | 0.8217 | 1 |
| MID | 1.0 | 0.7395 | 0.8731 | 0.8464 | 0.6186 | **0.9080** | 0.9070 | 0.8200 | 2 |
| MID | 10000 | 0.6989 | 0.8939 | 0.8549 | 0.6145 | 0.9071 | 0.9026 | 0.8197 | 3 |
| L-DP | 0.1 | 0.7407 | 0.8566 | 0.8334 | 0.6190 | 0.8957 | 0.8855 | 0.8084 | 4 |
| MID | 0.1 | 0.6937 | 0.8577 | 0.8249 | 0.6205 | 0.8964 | 0.8885 | 0.8076 | 5 |
| L-DP | 0.01 | 0.7244 | 0.8577 | 0.8310 | 0.6029 | 0.8920 | 0.8918 | 0.8044 | 6 |
| MID | 0.01 | **0.7411** | 0.8185 | 0.8031 | 0.6169 | 0.8504 | 0.8948 | 0.7913 | 7 |
| G-DP | 0.1 | 0.7392 | 0.8528 | 0.8301 | **0.6220** | 0.8913 | 0.8208 | 0.7910 | 8 |
| G-DP | 0.01 | 0.7350 | 0.8461 | 0.8238 | 0.6026 | 0.8813 | 0.8167 | 0.7811 | 9 |
| MID | 0.0 | 0.7335 | 0.8058 | 0.7913 | 0.6095 | 0.7862 | 0.8676 | 0.7636 | 10 |
| MID | 1e-06 | 0.7382 | 0.8048 | 0.7914 | 0.6125 | 0.7798 | 0.8680 | 0.7629 | 11 |
| MID | 1e-08 | 0.7382 | 0.8063 | 0.7927 | 0.6110 | 0.7714 | 0.8669 | 0.7605 | 12 |
| MID | 0.0001 | 0.6564 | 0.8037 | 0.7742 | 0.6168 | 0.7720 | 0.8653 | 0.7571 | 13 |
| L-DP | 0.001 | 0.7061 | 0.7720 | 0.7588 | 0.6022 | 0.7911 | 0.8725 | 0.7562 | 14 |
| dCor | 0.3 | 0.5901 | 0.7999 | 0.7579 | 0.6083 | 0.8277 | 0.8241 | 0.7545 | 15 |
| dCor | 0.1 | 0.5881 | 0.7241 | 0.6969 | 0.6046 | 0.8234 | 0.8203 | 0.7363 | 16 |
| G-DP | 0.001 | 0.7038 | 0.7564 | 0.7458 | 0.6021 | 0.7677 | 0.8200 | 0.7339 | 17 |
| dCor | 0.01 | 0.5858 | 0.6305 | 0.6216 | 0.6022 | 0.7999 | 0.8189 | 0.7106 | 18 |
| L-DP | 0.0001 | 0.5858 | 0.6727 | 0.6553 | 0.6020 | 0.6533 | 0.8238 | 0.6836 | 19 |
| G-DP | 0.0001 | 0.5858 | 0.6632 | 0.6477 | 0.6026 | 0.6640 | 0.8196 | 0.6835 | 20 |
| dCor | 0.0001 | 0.5858 | 0.6258 | 0.6178 | 0.6020 | 0.6645 | 0.8161 | 0.6751 | 21 |
| GS | 99.5 | 0.7388 | 0.8873 | **0.8576** | - | 0.7189 | 0.8454 | - | - |
| GS | 99.0 | 0.7388 | 0.8780 | 0.8502 | - | 0.7038 | 0.8313 | - | - |
| GS | 97.0 | 0.7388 | 0.7573 | 0.7536 | - | 0.6825 | 0.8305 | - | - |
| GS | 95.0 | 0.7388 | 0.7232 | 0.7263 | - | 0.6732 | 0.8268 | - | - |
| CAE | 1.0 | 0.5858 | 0.7960 | 0.7539 | - | 0.7632 | - | - | - |
| CAE | 0.1 | 0.5858 | 0.8144 | 0.7687 | - | 0.7124 | - | - | - |
| CAE | 0.5 | 0.5858 | 0.7937 | 0.7521 | - | 0.7280 | - | - | - |
| CAE | 0.0 | 0.5858 | 0.8218 | 0.7746 | - | 0.6837 | - | - | - |
| DCAE | 1.0 | 0.5858 | 0.9021 | 0.8389 | - | 0.7648 | - | - | - |
| DCAE | 0.5 | 0.5858 | 0.9039 | 0.8402 | - | 0.7396 | - | - | - |
| DCAE | 0.1 | 0.5858 | 0.8990 | 0.8364 | - | 0.7299 | - | - | - |
| DCAE | 0.0 | 0.5858 | **0.9094** | 0.8447 | - | 0.7111 | - | - | - |
| GPer | 10.0 | 0.5903 | 0.7016 | 0.6794 | - | 0.8031 | - | - | - |
| GPer | 1.0 | 0.7306 | 0.8097 | 0.7939 | - | 0.9045 | - | - | - |
| GPer | 0.1 | 0.7357 | 0.8546 | 0.8308 | - | 0.8926 | - | - | - |
| GPer | 0.01 | 0.7360 | 0.8525 | 0.8292 | - | 0.8857 | - | - | - |

Table 16: T-DCS and C-DCS for All Defenses [CIFAR10 dataset, aggVFL, FedSGD]

| Defense Name | Defense Parameter | $T\text{-}DCS_{LI_2}$ | $T\text{-}DCS_{LI_{10}}$ | $T\text{-}DCS_{LI}$ | $T\text{-}DCS_{FR}$ | $T\text{-}DCS_{TB}$ | $T\text{-}DCS_{NTB}$ | $C\text{-}DCS$ | Ranking |
|---|---|---|---|---|---|---|---|---|---|
| MID | 0.01 | 0.7232 | 0.9172 | 0.8784 | 0.6035 | 0.8942 | 0.9286 | 0.8262 | 1 |
| MID | 1.0 | 0.7260 | 0.9173 | **0.8791** | 0.6039 | 0.8931 | 0.9286 | 0.8262 | 2 |
| MID | 100 | 0.7270 | 0.9161 | 0.8783 | 0.6039 | 0.8931 | 0.9280 | 0.8258 | 3 |
| MID | 10000 | 0.7276 | 0.9159 | 0.8782 | 0.6016 | 0.8945 | 0.9284 | 0.8257 | 4 |
| MID | 0.1 | 0.7001 | **0.9175** | 0.8740 | 0.6016 | 0.8906 | 0.9284 | 0.8237 | 5 |
| MID | 1e-06 | 0.7132 | 0.8600 | 0.8306 | 0.6024 | 0.9123 | 0.9323 | 0.8194 | 6 |
| MID | 1e-08 | 0.7132 | 0.8594 | 0.8301 | 0.6023 | 0.9123 | 0.9313 | 0.8190 | 7 |
| MID | 0.0 | 0.7132 | 0.8552 | 0.8268 | 0.6023 | 0.9121 | 0.9289 | 0.8175 | 8 |
| MID | 0.0001 | 0.7118 | 0.8575 | 0.8283 | 0.6034 | 0.9061 | 0.9302 | 0.8170 | 9 |
| G-DP | 0.01 | 0.7316 | 0.8965 | 0.8635 | 0.6019 | 0.8809 | 0.9152 | 0.8154 | 10 |
| L-DP | 0.01 | 0.7365 | 0.8987 | 0.8663 | 0.6021 | 0.8751 | 0.9146 | 0.8145 | 11 |
| L-DP | 0.001 | 0.7041 | 0.8787 | 0.8437 | 0.6023 | 0.8892 | 0.9211 | 0.8141 | 12 |
| G-DP | 0.1 | 0.7324 | 0.8947 | 0.8622 | **0.6097** | 0.8670 | 0.9083 | 0.8118 | 13 |
| G-DP | 0.001 | 0.6704 | 0.8713 | 0.8311 | 0.6023 | 0.8879 | 0.9241 | 0.8114 | 14 |
| L-DP | 0.1 | 0.7353 | 0.8919 | 0.8606 | 0.6034 | 0.8696 | 0.9058 | 0.8098 | 15 |
| dCor | 0.3 | 0.6116 | 0.8109 | 0.7710 | 0.6012 | 0.8883 | 0.9245 | 0.7963 | 16 |
| dCor | 0.1 | 0.5879 | 0.7695 | 0.7332 | 0.6030 | 0.8962 | 0.9175 | 0.7875 | 17 |
| L-DP | 0.0001 | 0.5876 | 0.7611 | 0.7264 | 0.6023 | 0.9049 | 0.9083 | 0.7855 | 18 |
| G-DP | 0.0001 | 0.5868 | 0.7487 | 0.7163 | 0.6025 | 0.9046 | 0.8993 | 0.7807 | 19 |
| dCor | 0.0001 | 0.5858 | 0.6578 | 0.6434 | 0.6031 | 0.9162 | **0.9558** | 0.7796 | 20 |
| dCor | 0.01 | 0.5859 | 0.6839 | 0.6643 | 0.6042 | 0.9162 | 0.8982 | 0.7708 | 21 |
| GS | 99.5 | 0.7387 | 0.8956 | 0.8642 | - | 0.8877 | 0.9276 | - | - |
| GS | 99.0 | 0.7388 | 0.8810 | 0.8526 | - | 0.8888 | 0.9282 | - | - |
| GS | 97.0 | 0.7388 | 0.8367 | 0.8171 | - | 0.8882 | 0.9286 | - | - |
| GS | 95.0 | 0.7388 | 0.8049 | 0.7916 | - | 0.8895 | 0.9295 | - | - |
| CAE | 1.0 | 0.5918 | 0.8592 | 0.8057 | - | 0.8482 | - | - | - |
| CAE | 0.5 | 0.5860 | 0.8633 | 0.8079 | - | 0.8557 | - | - | - |
| CAE | 0.1 | 0.5858 | 0.8498 | 0.7970 | - | 0.9179 | - | - | - |
| CAE | 0.0 | 0.5858 | 0.8419 | 0.7907 | - | **0.9214** | - | - | - |
| DCAE | 1.0 | 0.6080 | 0.8678 | 0.8159 | - | 0.8464 | - | - | - |
| DCAE | 0.5 | 0.6131 | 0.8780 | 0.8250 | - | 0.8556 | - | - | - |
| DCAE | 0.1 | 0.6090 | 0.8977 | 0.8400 | - | 0.9039 | - | - | - |
| DCAE | 0.0 | 0.6090 | 0.9038 | 0.8449 | - | 0.9066 | - | - | - |
| GPer | 10.0 | 0.6422 | 0.8082 | 0.7750 | - | 0.8937 | - | - | - |
| GPer | 1.0 | 0.6976 | 0.8913 | 0.8526 | - | 0.8838 | - | - | - |
| GPer | 0.1 | 0.7385 | 0.8946 | 0.8634 | - | 0.8715 | - | - | - |
| GPer | 0.01 | **0.7392** | 0.8931 | 0.8623 | - | 0.8700 | - | - | - |

Table 17: T-DCS and C-DCS for All Defenses [NUSWIDE dataset, aggVFL, FedSGD]

| Defense Name | Defense Parameter | $T\text{-}DCS_{LI_2}$ | $T\text{-}DCS_{LI_5}$ | $T\text{-}DCS_{LI}$ | $T\text{-}DCS_{FR}$ | $T\text{-}DCS_{TB}$ | $T\text{-}DCS_{NTB}$ | $C\text{-}DCS$ | Ranking |
|---|---|---|---|---|---|---|---|---|---|
| MID | 10000 | 0.7358 | 0.8559 | **0.8159** | 0.5833 | **0.7333** | 0.8707 | 0.7508 | 1 |
| MID | 1.0 | 0.7476 | 0.8472 | 0.8140 | 0.5833 | 0.7331 | 0.8700 | 0.7501 | 2 |
| MID | 100 | 0.7320 | 0.8536 | 0.8130 | 0.5833 | 0.7326 | **0.8711** | 0.7500 | 3 |
| G-DP | 0.1 | 0.7375 | 0.8262 | 0.7966 | 0.5863 | 0.7282 | 0.8675 | 0.7447 | 4 |
| L-DP | 0.1 | 0.7389 | 0.8177 | 0.7915 | 0.5863 | 0.7258 | 0.8603 | 0.7410 | 5 |
| MID | 0.1 | 0.7516 | 0.8259 | 0.8011 | 0.5833 | 0.7172 | 0.8563 | 0.7395 | 6 |
| MID | 0.01 | 0.7280 | 0.8092 | 0.7822 | 0.5844 | 0.7151 | 0.8627 | 0.7361 | 7 |
| MID | 0.0001 | 0.7144 | 0.8097 | 0.7779 | 0.5856 | 0.7040 | 0.8680 | 0.7339 | 8 |
| dCor | 0.3 | **0.7641** | 0.8411 | 0.8155 | 0.5834 | 0.7289 | 0.8051 | 0.7332 | 9 |
| G-DP | 0.01 | 0.7391 | 0.7600 | 0.7530 | 0.5863 | 0.7061 | 0.8549 | 0.7251 | 10 |
| L-DP | 0.01 | 0.7395 | 0.7525 | 0.7482 | 0.5863 | 0.7148 | 0.8485 | 0.7244 | 11 |
| MID | 1e-06 | 0.7022 | 0.8201 | 0.7808 | 0.5860 | 0.6880 | 0.8408 | 0.7239 | 12 |
| dCor | 0.1 | 0.7442 | 0.7617 | 0.7559 | 0.5841 | 0.7259 | 0.8279 | 0.7234 | 13 |
| MID | 1e-08 | 0.7066 | 0.8147 | 0.7787 | 0.5862 | 0.6593 | 0.8410 | 0.7163 | 14 |
| MID | 0.0 | 0.6599 | 0.8097 | 0.7598 | 0.5862 | 0.6590 | 0.8414 | 0.7116 | 15 |
| L-DP | 0.001 | 0.7291 | 0.7234 | 0.7253 | 0.5863 | 0.6424 | 0.8329 | 0.6967 | 16 |
| G-DP | 0.001 | 0.7175 | 0.7237 | 0.7216 | 0.5863 | 0.6379 | 0.8334 | 0.6948 | 17 |
| dCor | 0.01 | 0.7445 | 0.7021 | 0.7162 | 0.5863 | 0.6336 | 0.8295 | 0.6914 | 18 |
| L-DP | 0.0001 | 0.6783 | 0.6470 | 0.6574 | 0.5863 | 0.6313 | 0.8293 | 0.6761 | 19 |
| G-DP | 0.0001 | 0.6495 | 0.6381 | 0.6419 | 0.5863 | 0.6309 | 0.8290 | 0.6720 | 20 |
| dCor | 0.0001 | 0.6496 | 0.6340 | 0.6392 | **0.5864** | 0.6307 | 0.8287 | 0.6712 | 21 |
| GS | 99.5 | 0.7381 | 0.8142 | 0.7888 | - | 0.6456 | 0.8415 | - | - |
| GS | 99.0 | 0.7404 | 0.8060 | 0.7841 | - | 0.6415 | 0.8408 | - | - |
| GS | 97.0 | 0.7414 | 0.7672 | 0.7586 | - | 0.6376 | 0.8392 | - | - |
| GS | 95.0 | 0.7423 | 0.7399 | 0.7407 | - | 0.6375 | 0.8385 | - | - |
| CAE | 1.0 | 0.6863 | 0.7822 | 0.7502 | - | 0.6830 | - | - | - |
| CAE | 0.5 | 0.6808 | 0.7848 | 0.7501 | - | 0.6733 | - | - | - |
| CAE | 0.1 | 0.6808 | 0.8249 | 0.7768 | - | 0.6734 | - | - | - |
| CAE | 0.0 | 0.6808 | 0.8212 | 0.7744 | - | 0.6807 | - | - | - |
| DCAE | 1.0 | 0.6716 | 0.8156 | 0.7676 | - | 0.6771 | - | - | - |
| DCAE | 0.5 | 0.6672 | 0.8108 | 0.7629 | - | 0.6668 | - | - | - |
| DCAE | 0.1 | 0.6669 | 0.8651 | 0.7991 | - | 0.6746 | - | - | - |
| DCAE | 0.0 | 0.6669 | **0.8660** | 0.7996 | - | 0.6816 | - | - | - |
| GPer | 10.0 | 0.6877 | 0.6722 | 0.6773 | - | 0.6222 | - | - | - |
| GPer | 1.0 | 0.7230 | 0.7460 | 0.7383 | - | 0.6315 | - | - | - |
| GPer | 0.1 | 0.7395 | 0.8007 | 0.7803 | - | 0.7042 | - | - | - |
| GPer | 0.01 | 0.7386 | 0.8412 | 0.8070 | - | 0.7193 | - | - | - |

Table 18: T-DCS and C-DCS for All Defenses [MNIST dataset, splitVFL, FedSGD]

| Defense Name | Defense Parameter | $T\text{-}DCS_{LI_2}$ | $T\text{-}DCS_{LI_{10}}$ | $T\text{-}DCS_{LI}$ | $T\text{-}DCS_{FR}$ | $T\text{-}DCS_{TB}$ | $T\text{-}DCS_{NTB}$ | $C\text{-}DCS$ | Ranking |
|---|---|---|---|---|---|---|---|---|---|
| MID | 1.0 | 0.7380 | 0.8846 | 0.8553 | 0.6065 | 0.9051 | 0.9237 | 0.8226 | 1 |
| L-DP | 0.1 | 0.7398 | 0.8700 | 0.8439 | 0.6156 | 0.9092 | 0.9203 | 0.8222 | 2 |
| G-DP | 0.1 | 0.7350 | 0.8654 | 0.8393 | 0.6139 | 0.9058 | **0.9271** | 0.8215 | 3 |
| MID | 10000 | 0.7372 | **0.8921** | **0.8611** | 0.6064 | 0.9071 | 0.9062 | 0.8202 | 4 |
| MID | 100 | 0.7371 | 0.8896 | 0.8591 | 0.6082 | 0.9085 | 0.9034 | 0.8198 | 5 |
| L-DP | 0.01 | 0.7381 | 0.8603 | 0.8359 | 0.6026 | 0.9099 | 0.9147 | 0.8158 | 6 |
| MID | 0.1 | 0.7398 | 0.8652 | 0.8401 | **0.6241** | **0.9113** | 0.8871 | 0.8156 | 7 |
| G-DP | 0.01 | 0.7393 | 0.8562 | 0.8328 | 0.6031 | 0.9060 | 0.9170 | 0.8147 | 8 |
| dCor | 0.3 | 0.5934 | 0.8876 | 0.8288 | 0.6194 | 0.9072 | 0.8732 | 0.8071 | 9 |
| L-DP | 0.001 | 0.7189 | 0.8365 | 0.8130 | 0.6038 | 0.8682 | 0.8663 | 0.7878 | 10 |
| MID | 0.01 | 0.7364 | 0.8455 | 0.8237 | 0.6140 | 0.8421 | 0.8536 | 0.7834 | 11 |
| G-DP | 0.001 | **0.7404** | 0.8269 | 0.8096 | 0.6041 | 0.8109 | 0.8775 | 0.7755 | 12 |
| dCor | 0.1 | 0.5983 | 0.8519 | 0.8012 | 0.6133 | 0.8519 | 0.8212 | 0.7719 | 13 |
| MID | 1e-06 | 0.7388 | 0.8140 | 0.7990 | 0.6173 | 0.7641 | 0.8394 | 0.7549 | 14 |
| MID | 1e-08 | 0.7388 | 0.8158 | 0.8004 | 0.6135 | 0.7615 | 0.8421 | 0.7544 | 15 |
| MID | 0.0 | 0.7388 | 0.8147 | 0.7995 | 0.6146 | 0.7524 | 0.8456 | 0.7530 | 16 |
| dCor | 0.01 | 0.5870 | 0.8020 | 0.7590 | 0.6048 | 0.8111 | 0.8174 | 0.7481 | 17 |
| MID | 0.0001 | 0.7388 | 0.8137 | 0.7988 | 0.6182 | 0.7324 | 0.8312 | 0.7451 | 18 |
| L-DP | 0.0001 | 0.6472 | 0.8041 | 0.7727 | 0.6046 | 0.7699 | 0.8268 | 0.7435 | 19 |
| G-DP | 0.0001 | 0.6253 | 0.7972 | 0.7628 | 0.6036 | 0.7586 | 0.8212 | 0.7365 | 20 |
| dCor | 0.0001 | 0.5858 | 0.7906 | 0.7496 | 0.6053 | 0.7538 | 0.8164 | 0.7313 | 21 |
| GS | 95.0 | 0.7388 | 0.8080 | 0.7941 | - | 0.7380 | 0.8345 | - | - |
| GS | 97.0 | 0.7388 | 0.8133 | 0.7984 | - | 0.7355 | 0.8428 | - | - |
| GS | 99.0 | 0.7388 | 0.8334 | 0.8145 | - | 0.7590 | 0.8558 | - | - |
| GS | 99.5 | 0.7388 | 0.8487 | 0.8267 | - | 0.7872 | 0.8636 | - | - |
| CAE | 1.0 | 0.5858 | 0.8252 | 0.7773 | - | 0.7741 | - | - | - |
| CAE | 0.5 | 0.5858 | 0.8282 | 0.7797 | - | 0.7439 | - | - | - |
| CAE | 0.1 | 0.5858 | 0.8365 | 0.7863 | - | 0.7535 | - | - | - |
| CAE | 0.0 | 0.5858 | 0.8410 | 0.7899 | - | 0.7574 | - | - | - |
| DCAE | 1.0 | 0.5858 | 0.7824 | 0.7431 | - | 0.7882 | - | - | - |
| DCAE | 0.5 | 0.5858 | 0.7862 | 0.7461 | - | 0.7882 | - | - | - |
| DCAE | 0.1 | 0.5858 | 0.7773 | 0.7390 | - | 0.7882 | - | - | - |
| DCAE | 0.0 | 0.5858 | 0.7758 | 0.7378 | - | 0.7882 | - | - | - |
| GPer | 10.0 | 0.5916 | 0.8037 | 0.7613 | - | 0.8490 | - | - | - |
| GPer | 1.0 | 0.7295 | 0.8460 | 0.8227 | - | 0.9064 | - | - | - |
| GPer | 0.1 | 0.7369 | 0.8632 | 0.8379 | - | 0.9039 | - | - | - |
| GPer | 0.01 | 0.7365 | 0.8712 | 0.8442 | - | 0.9025 | - | - | - |

Table 19: T-DCS and C-DCS for All Defenses [MNIST dataset, aggVFL, **FedBCD(Q=5)**]

| Defense Name | Defense Parameter | $T\text{-}DCS_{LI_2}$ | $T\text{-}DCS_{LI_{10}}$ | $T\text{-}DCS_{LI}$ | $T\text{-}DCS_{FR}$ | $T\text{-}DCS_{TB}$ | $T\text{-}DCS_{NTB}$ | $C\text{-}DCS$ | Ranking |
|---|---|---|---|---|---|---|---|---|---|
| MID | 1.0 | 0.6707 | 0.8794 | 0.8376 | 0.6166 | 0.9034 | 0.9397 | 0.8243 | 1 |
| MID | 10000 | 0.6917 | 0.8729 | 0.8366 | 0.6133 | 0.9034 | **0.9417** | 0.8238 | 2 |
| MID | 100 | 0.6908 | 0.8750 | 0.8381 | 0.6087 | **0.9039** | 0.9347 | 0.8214 | 3 |
| MID | 0.1 | 0.6121 | 0.8730 | 0.8208 | 0.6199 | 0.9030 | 0.9413 | 0.8212 | 4 |
| L-DP | 0.1 | 0.7403 | 0.8369 | 0.8176 | **0.6525** | 0.8669 | 0.8506 | 0.7969 | 5 |
| G-DP | 0.1 | 0.7385 | 0.8402 | 0.8199 | 0.6262 | 0.8704 | 0.8532 | 0.7924 | 6 |
| G-DP | 0.01 | **0.7404** | 0.8376 | 0.8182 | 0.6037 | 0.8976 | 0.8464 | 0.7915 | 7 |
| L-DP | 0.01 | 0.7387 | 0.8331 | 0.8142 | 0.6055 | 0.8959 | 0.8472 | 0.7907 | 8 |
| MID | 0.01 | 0.6119 | 0.8357 | 0.7910 | 0.6166 | 0.8075 | 0.9118 | 0.7817 | 9 |
| L-DP | 0.001 | 0.6821 | 0.7915 | 0.7696 | 0.6033 | 0.8447 | 0.8668 | 0.7711 | 10 |
| G-DP | 0.001 | 0.7132 | 0.7836 | 0.7696 | 0.6051 | 0.8032 | 0.8527 | 0.7576 | 11 |
| dCor | 0.3 | 0.5900 | 0.8329 | 0.7843 | 0.6071 | 0.8000 | 0.8154 | 0.7517 | 12 |
| MID | 1e-06 | 0.6156 | 0.8089 | 0.7702 | 0.6144 | 0.7441 | 0.8727 | 0.7504 | 13 |
| MID | 0.0001 | 0.6155 | 0.8110 | 0.7719 | 0.6217 | 0.7166 | 0.8753 | 0.7464 | 14 |
| MID | 0.0 | 0.6156 | 0.8072 | 0.7689 | 0.6176 | 0.7396 | 0.8542 | 0.7451 | 15 |
| MID | 1e-08 | 0.6156 | 0.8092 | 0.7705 | 0.6224 | 0.7237 | 0.8522 | 0.7422 | 16 |
| dCor | 0.1 | 0.5960 | 0.7971 | 0.7569 | 0.6077 | 0.6955 | 0.8115 | 0.7179 | 17 |
| L-DP | 0.0001 | 0.5858 | 0.7246 | 0.6969 | 0.6044 | 0.7083 | 0.8241 | 0.7084 | 18 |
| G-DP | 0.0001 | 0.5858 | 0.7194 | 0.6927 | 0.6034 | 0.6963 | 0.8196 | 0.7030 | 19 |
| dCor | 0.01 | 0.5858 | 0.6900 | 0.6691 | 0.6052 | 0.6894 | 0.8121 | 0.6940 | 20 |
| dCor | 0.0001 | 0.5858 | 0.6763 | 0.6582 | 0.6055 | 0.6714 | 0.8086 | 0.6859 | 21 |
| GS | 99.5 | 0.7388 | 0.9005 | 0.8682 | - | 0.7086 | 0.8587 | - | - |
| GS | 99.0 | 0.7388 | 0.8896 | 0.8594 | - | 0.7093 | 0.8506 | - | - |
| GS | 97.0 | 0.7388 | 0.7718 | 0.7652 | - | 0.6683 | 0.8483 | - | - |
| GS | 95.0 | 0.7388 | 0.7458 | 0.7444 | - | 0.6585 | 0.8359 | - | - |
| CAE | 1.0 | 0.5858 | 0.9319 | 0.8627 | - | 0.7450 | - | - | - |
| CAE | 0.5 | 0.5858 | 0.9367 | 0.8665 | - | 0.8109 | - | - | - |
| CAE | 0.1 | 0.5858 | 0.9664 | 0.8903 | - | 0.6757 | - | - | - |
| CAE | 0.0 | 0.5858 | **0.9681** | **0.8917** | - | 0.6724 | - | - | - |
| DCAE | 1.0 | 0.5858 | 0.8734 | 0.8159 | - | 0.7311 | - | - | - |
| DCAE | 0.5 | 0.5858 | 0.8776 | 0.8192 | - | 0.8041 | - | - | - |
| DCAE | 0.1 | 0.5858 | 0.8867 | 0.8265 | - | 0.6619 | - | - | - |
| DCAE | 0.0 | 0.5858 | 0.8919 | 0.8307 | - | 0.6896 | - | - | - |
| GPer | 10.0 | 0.5901 | 0.7562 | 0.7230 | - | 0.7515 | - | - | - |
| GPer | 1.0 | 0.7317 | 0.8216 | 0.8036 | - | 0.8848 | - | - | - |
| GPer | 0.1 | 0.7341 | 0.8398 | 0.8186 | - | 0.8829 | - | - | - |
| GPer | 0.01 | 0.7366 | 0.8381 | 0.8178 | - | 0.8683 | - | - | - |

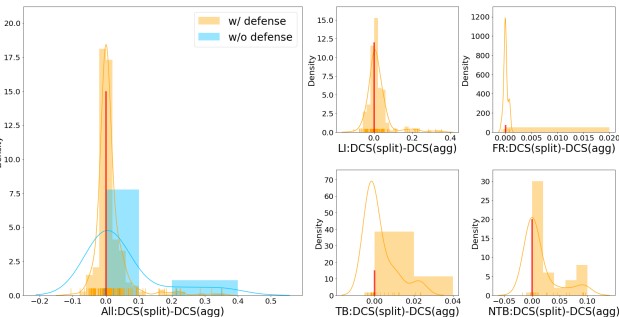

Figure 17: DCS gap Distribution, y-axis represents density [NUSWIDE dataset, splitVFL/aggVFL, FedSGD]

### I.2.4 ADDITIONAL RESULTS ON SPLITVFL AND AGGVFL COMPARISON

We further expand the discussion of the conclusions on the comparison between splitVFL and aggVFL as well as the comparison between FedBCD and FedSGD given in Sec. 6.2 here due to space limitations.

The conclusion that *when no defense is applied, a splitVFL system is less vulnerable to attacks than aggVFL* is also evident by the results under NUSWIDE dataset, with all the black square points in Fig. 18 appearing above or close to the red horizontal line at a value of $0.0$ and the blue histograms appearing mostly at the right of the vertical line at a value of $0.0$ in Fig. 17, indicating a positive DCS gap. This shows that a global trainable model can be beneficial to model robustness and safety. Specifically, the DCS gap is pronounced for attacks that directly exploit the gradients, i.e., DLI and BLI attacks.

The conclusion that *splitVFL has an overall positive effect on boosting defense performance against attacks* is also shown from Figs. 5 and 17 in which most DCS gaps for defenses are positive, especially

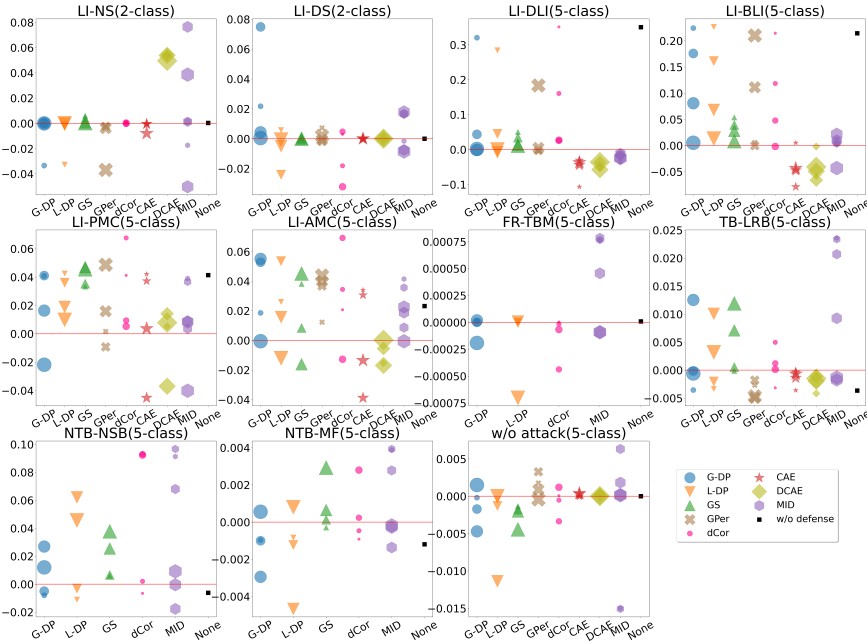

Figure 18: DCS gap for each attack-defense point [NUSWIDE dataset, splitVFL/aggVFL, FedSGD]

for LI, FR and TB attacks. This implies that splitVFL architecture exhibits greater robustness against potential attacks. However, not all defenses are enhanced in splitVFL setting. For example, in Fig. 16, MID results in minor negative DCS gaps in several attacks like GRN and NSB, so is GS in MC attacks.

The DCS ranking under splitVFL setting with FedSGD communication protocol using MNIST dataset is also included in Tab. 18, which is quite similar to that in Tab. 15 under aggVFL setting with FedSGD communication protocol using MNIST dataset. This indicates the robustness of our DCS evaluation metrics as well as the inner consistency of defense abilities under different settings.

### I.2.5 ADDITIONAL RESULTS ON FEDBCD AND FEDSGD COMPARISON

The conclusion given in Sec. 6.2 that *a system is less vulnerable to attacks under FedBCD setting when no defense method is applied* is also evident from Fig. 19, with all the black square points, except the one for MF attack, appear on or above the red horizontal line at a value of $0.0$. Also, the vulnerability of a system under FedBCD setting to attacks differs between different attacks. Specifically, as shown in Figs. 6 and 19, evaluating with MNIST dataset, VFL trained with FedBCD is much less vulnerable to BLI attack, which exploits batch-level gradient to recover labels. This is because gradients from earlier epochs of an un-trained model shared under FedSGD reveal more information about labels compared to FedBCD which only shares gradients every $Q > 1$ iterations. Similar results can be seen from Figs. 20 and 21 that compares the DCS between FedSGD and FedBCD under aggVFL setting with NUSWIDE dataset.

The DCS ranking using FedBCD communication protocol under aggVFL setting using MNIST dataset is also included in Tab. 19, which is quite similar to that in Tab. 15 using FedSGD communication protocol under aggVFL setting using MNIST dataset. This also indicates the robustness of our DCS evaluation metrics as well as the inner consistency of defense abilities under different settings.

### I.2.6 CONSISTENCY OF C-DCS RANKING.

We conduct a comparative analysis of the C-DCS rankings that are presented in Tabs. 15 to 19 across multiple datasets, communication protocols, and model partition strategies. The mean and standard

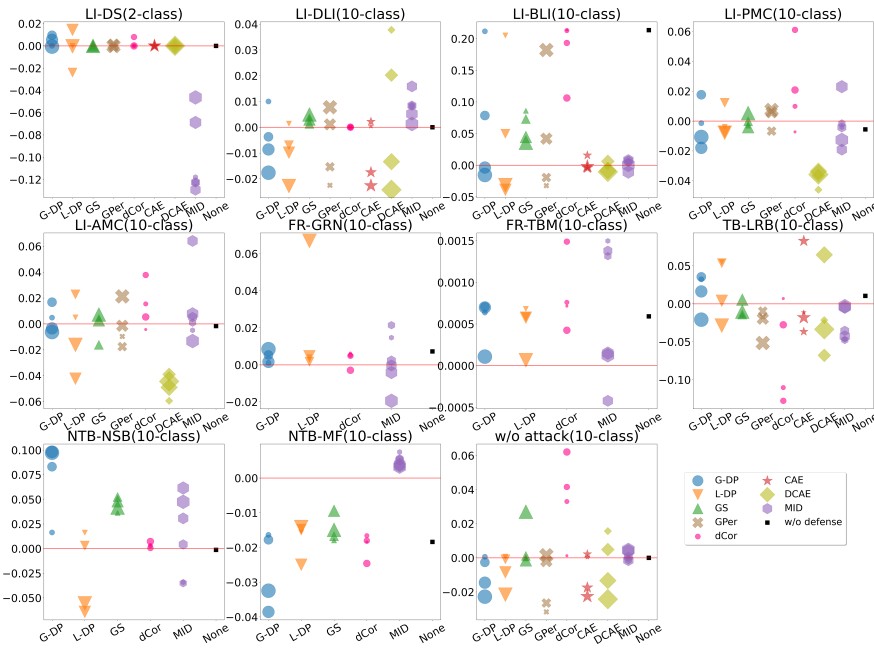

Figure 19: DCS gap for each attack-defense point [MNIST dataset, aggVFL, FedBCD/FedSGD]

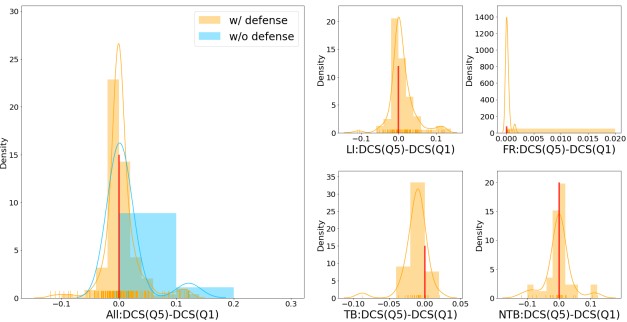

Figure 20: DCS gap Distribution, y-axis represents density [NUSWIDE dataset, aggVFL, Fed-BCD/FedSGD]

deviation of the 5 rankings for each defense are calculated and visualized in Fig. 22. Remarkably, despite the diversity in datasets, communication methods, and model partitioning, the variations in the rankings remain at a low level. This suggests that the C-DCS rankings are generally consistent across various datasets, communication protocol and model partition settings, and that the relative performance of different defense methods is relatively stable under different datasets and settings.

## J  SOCIAL IMPACT OF THE WORK

Our work introduces an extensible and lightweight VFL platform for research which will for sure facilitate the research considering VFL. Moreover, our work encourages not only the development of stronger defense methods, but also the development of new attacks for practical VFL scenarios. These attacks can be used for either benign or malicious purpose. Developing stronger defense or exploring related regulations and laws are possible approaches for alleviating the potential negative impacts.

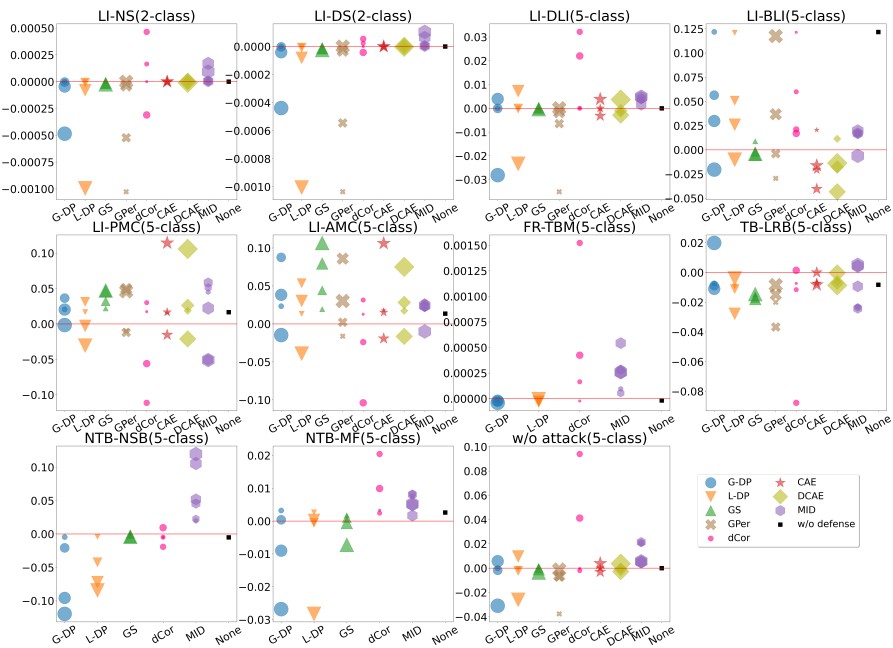

Figure 21: DCS gap for each attack-defense point [NUSWIDE dataset, aggVFL, FedBCD/FedSGD]

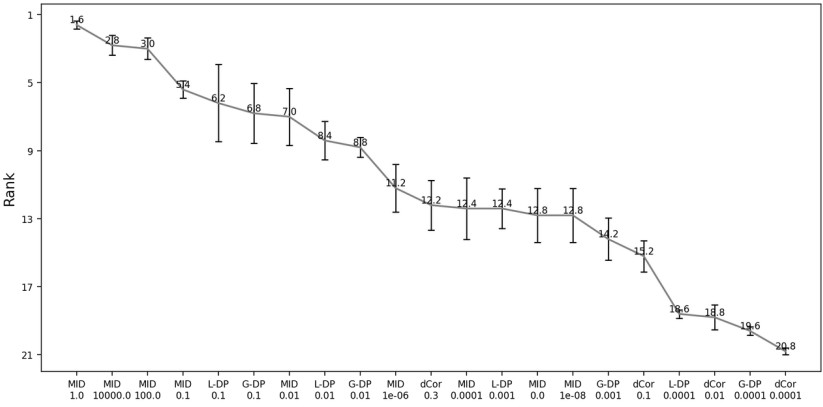

Figure 22: C-DCS ranking comparison across 3 datasets, 2 communication protocols and 2 dataset partition strategies presented in Tabs. 15 to 19.

