# OpenReview forum: "VFLAIR: A Research Library and Benchmark for Vertical Federated Learning"
_ICLR.cc/2024/Conference — ICLR 2024 poster_

### Official Review · Reviewer_Gb2U · 2023-10-28

**Soundness:** 3 good
**Presentation:** 4 excellent
**Contribution:** 4 excellent
**Rating:** 8
**Confidence:** 4

**Summary:**

This paper aims to develop a lightweight vertical federated learning (VFL) platform (VFLAIR) framework consisting of multiple model partitions, communication protocols, and attack and defense algorithms using datasets of different modalities. Under this platform, the unified evaluation metrics and benchmark defense performance with various attacks are introduced,  which sheds light on choosing defense techniques in practical deployment.
Although this paper summarizes most of the VFL settings, some advanced model partition and communication protocol algorithms are missed. In this way, I suggest authors should add these missed algorithms. In addition, I think some evaluations of this platform should be developed to show how it is lightweight. Some recommendation tasks should be considered, such as Criteo, Avazu, etc.

**Strengths:**

This paper can provide a VFL platform for researchers to evaluate the performance and efficiency of their proposed algorithms. This platform from five aspects, i.e., model partitions, communication protocols, attacks, defenses, and dataset modalities, which are very significant for VFL studies. The strengths of this paper are as follows:
1. Most of the VFL settings, e.g., model partitions, communication protocols, attacks, defenses, and dataset modalities, are included in this platform.
2. This platform provides unified evaluation metrics and benchmark defense performance with various attacks,  which can guide the selection of defense techniques in practical deployment.
3. The analysis of experimental results is sufficient and insightful.

**Weaknesses:**

Some weaknesses are shown as follows:
1. Although this paper summarizes most of the VFL settings, some advanced model partition and communication protocol algorithms are missed.  In this way, I suggest authors should add these missed algorithms, such as quantization, federated graph neural networks, etc.
2. I think some evaluations of this platform should be developed to show how it is lightweight.
3. VFL is usually adopted in recommendation tasks instead of image classification. The authors apply too many image classification datasets in the experiments.
4. As discussed in limitations, the cryptographic techniques that are significant, are not included in this library.
5. The communication and computation efficiencies are also very important. The metrics should include the evaluation of the communication and computation efficiencies.

**Questions:**

1. Please add some advanced model partition and communication protocol algorithms, such as quantization, federated graph neural networks, etc.
2. Add some compassion and discussion on how this platform is lightweight.
3. The authors apply too many image classification datasets in the experiments. Some recommendation tasks should be considered, such as Criteo, Avazu, etc.
4. As discussed in limitations, the cryptographic techniques that are significant, are not included in this library.
5. The communication and computation efficiencies are also very important. The metrics should include the evaluation of the communication and computation efficiencies.

---

> ### Author Response · Authors · 2023-11-18
> **Responses to Questions and Weaknesses from R-Gb2U**
>
> ### **Response to Weakness 1 and Question 1**
>
> Thank you for your constructive suggestions. We have included 3 new communication protocols, including Quantizate [1], Top-k [1] and CELU-VFL [2]. We also added FedGNN to benchmark its performance on Cora dataset. Details on these experimental settings including model partition, hyperparameters are shown in Tab. 10 to 12 in Appendix H. Results are all included in Tab. 6 for communication protocols and in Tab. 7 for Cora dataset with FedGNN, both in Section 6.1. For quicker access, please refer to our reply in *"Response to Weakness 4 and Question 3 for Reviewer z5tx"* (communication protocols) and *“Response to Weakness 3 and Question 3 for Reviewer z5tx”* (Cora dataset with FedGNN, Tab(1) in the response). We do not show them here again to avoid repeatitiveness and redundancy.
>
> ### **Response to Weakness 2 and Question 2**
>
> To demonstrate that our platform is lightweight, we compare it with FATE, one of most widely used FL platform supporting a broad range of VFL functionalities, on their system requirements for deployment.  According to FATE ([link](https://github.com/FederatedAI/FATE/blob/master/deploy/standalone-deploy/README.md)), deploying a stand-alone version of the FATE framework requires at least a 8 core CPU with 16G memory and 500G hard disk and the downloaded docker package for deployment is of size 4.92G for version 1.7.1.1. However, for our VFLAIR, a 1 core CPU with less than 4G memory and less than 4.0G hard disk is required for installation.
>
> (4) Comparision between FATE and VFLAIR
> |                                     |       CPU      |              memory              | Installation required hard disk |
> |:-----------------------------------:|:--------------:|:--------------------------------:|:-------------------------------:|
> | FATE (stand-alone, version 1.7.1.1) |     8 core     |                16G               |              4.92G              |
> |                VFLAIR               |     1 core     |                4G                |              $<$4G              |
>
> ### **Response to Weakness 3 and Question 3**
> Thank you for your suggestions. We have added Criteo and Avazu into our experiments, and results are included in Tab. 7 in Section 6.1. For quicker access, please refer to our reply in *“Response to Weakness 3 and Question 3 for Reviewer z5tx”* (Tab(1) in the response). We do not show them again here to avoid repeatitiveness and redundancy.
>
> ### **Response to Weakness 4 and Question 4**
>
> Thanks for the suggestion. We prioritized benchmarking non-cryptographic defense methods because they are emerging more rapidly in recent years due to their capability of defending various types of atttacks, whereas cryptographic techniques typically protect data in-transit during training, can not defend backdoor attacks or inference-time attacks effecitively while incurring significant computation and communiction cost to run. The benchmark of cryptographic techniques focus more on efficiency. Nevertheless, we think that the combination of cryptographic and noncryptographic methods would be an interesting research direction and potential solution for practical deployment.
>
> For VFLAIR, we have already included one of most commonly used cryptographic techniques, i.e. Paillier Encryption, in tree-based VFL and evaluated its impact on model performance and computation efficiency in Table. 5 in Section 6. We are currently implementing Encryption techniques to NN-based VFL and also plan to add more advanced privacy-preserving cryptographic methods to our VFLAIR in the near future.
>
> ### **Response to Weakness 5 and Question 5**
> Thank you for your suggestions. We have now formally added **communication rounds (\#Rounds)** and **amount of exchanged inforation each round (Amount)** for the evaluation of communication efficiency, as well as **Execution Time (Exec. Time)** for evaluating computation efficiency in Section 5.3 Evaluation Metrics. Further we applied these metrics in our results in Section 6.1 when comparing multiple communication protocols in Tab. 6, and when comparing the computation efficiency in Tab. 5.
>
> **Reference**
>
> [1] Castiglia, Timothy J., et al. "Compressed-vfl: Communication-efficient learning with vertically partitioned data." International Conference on Machine Learning. PMLR, 2022.
>
> [2] Fu, Fangcheng, et al. "Towards communication-efficient vertical federated learning training via cache-enabled local updates." arXiv preprint arXiv:2207.14628 (2022).

---

> ### Author Response · Authors · 2023-11-22
> **We are anxiously looking forward to your replies to our responses**
>
> Dear reviewer, do our responses to your questions as well as the updated paper address your concerns? We have added federated graph neural network as advanced model (Tab. 7 in Section 6); quantization, Top-k compression as well as CELU-VFL for additional communication protocols (Tab. 6 in Section 6); recommandation datasets including Criteo and Avazu (Tab. 7 in Section 6); as well as more communication and computation effectiveness evaluation metrics to our VFLAIR platform and benchmark experiments (in Section 6) according to your reviews. We have also added comparition to FATE to show that our platform is lightweight (in Appendix D). We are currently working on supporting Paillier Encryption in Linear Regression models in our platform. We hope the above have solved your concerns and we are still looking forward to your further replies. We will remain open and available if there is any further concerns.

---

### Official Review · Reviewer_1nCt · 2023-10-31

**Soundness:** 2 fair
**Presentation:** 2 fair
**Contribution:** 1 poor
**Rating:** 3
**Confidence:** 4

**Summary:**

This paper proposed a benchmarking framework for vertical federated learning. It proposes new evaluation metrics such as defence capability score (DCS)

**Strengths:**

This paper aims to propose a framework that can provide universal benchmarking solution for vertical federated learning.

The literature review is commendable, especially on the attack and defence part.

**Weaknesses:**

- The paper predominantly centers on evaluating attacks and defence strategies. But the paper title implies a broader scope – VFL in its entirety. The paper title could be more specific to align with the focus of the paper.

- A notable contribution is the introduction of new evaluation metrics such as defence capability score (DCS). However, the experiments did not validate the effectiveness of the proposed metrics. What is the evidence that shows that the proposed metrics indeed work, representing the real ability of the evaluated algorithms?

- The paper claims to “implement basic VFL training and evaluation flow under multiple model partition, communication protocols and attacks and defences algorithms using datasets of different modality”. But it lacks a clear exposition of the workflow. What is the training and evaluation flow in VFLAIR and how does the workflow facilitate the benchmarking?

- Following on the above point, the evaluation section is not organised systematically according to model partition, communication protocols, and attack and defence algorithms. The current evaluation section only amounts to a compilation of experimental results, which
requires a more structured, systematic and coherent organization.

There is only one paragraph in related work. There is no need to employ a bullet point at the beginning

**Questions:**

See the weakness part

---

> ### Author Response · Authors · 2023-11-18
> **Responses to Questions and Weaknesses from R-1nCt**
>
> ### **Response to Weakness 1**
>
> Thank you for the suggestion. As a **research library** designed to facilitate research development in the VFL, our VFLAIR incorporated all the essential components of a VFL system, including 2 model partition strategies, 5 communication protocols (3 added during discussion), 1 encryption protocol, 13 datasets (4 added during discussion) of different modality and data partition methods, 29 model architectures, in addition to 11 attacks and 8 defenses. In addition, our VFLAIR can be easily used and extended to perform evaluations on various aspects of a VFL system, including model performance, computation and communication efficiency under a broad range of settings.
>
> Therefore, we consider our VFLAIR a comprehensive VFL library and will continue to add more state-of-art privacy-preserving and communication-efficient methods in the future. As for our benchmark evaluations, we have further refined and completed new evaluations on model performance of new datasets (Tab. 7 in Section 6.1), communication and computation efficiency of various communication strategies (Tab. 6 in Section 6.1). Therefore, although comprehensive evaluation of attack and defense is an outstanding feature of our benchmark, we think other aspects should not be overlooked. So we would prefer to **keep our title unchanged**.
>
> ### **Response to Weakness 2**
>
> We validate the effectiveness of our proposed DCS metrics from the following 3 aspects.
>
> - **Robustness.**  The results of the C-DCS rankings are generally consistent across all datasets (MNIST, CIAFR10 and NUSWIDE) in Tabs. 14 to 16 under the same setting. Also, the C-DCS ranking of the defense methods are still generally consistent under different model partition setting and communication protocols in Tabs. 14, 17 and 18. Below we plot the mean and standard derivation (std) of the ranking of each defense in the above 5 ranking tables with different defense hyper-parameter shown in [this figure (link)](https://anonymous.4open.science/r/VFLAIR_discussion/dcs_roubustness.png). As shown in this figure,  the ranking is relatively stable and the std is small across multiple settings for all the defenses. Finally, as shown in Figs. 4, 14 and 15, the C-DCS ranking is also generally stable with the change of $\beta$, especially when $\beta \leq 0.7$. All these demonstrate the good stability and credibility of our proposed C-DCS metrics.
> - **Sensitivity.** We demonstrate that the changes in DCS value is a good indicator of relative defense capability with the following evidence: (1) From Tabs. 14 to 16, we show that T-DCS$_{FR}$ (T-DCS for feature reconstruction attacks) values are much lower than the T-DCS values of other attack types, indicating inferior defense capabilities under feature reconstruction attacks. This phenomenon is consistent with the fact that feature reconstruction attacks, especially TBM which uses auxiliary data for reconstruction, result in much stronger attack capabilities (shown in Figs. 3, 11 and 12). Here we also show the visualization of the reconstructed images of TBM under different defense methods in [this figure (link)](https://anonymous.4open.science/r/VFLAIR_discussion/5ressfl.png). It is apparent that although the reconstructed images are noisy, the contour can still be seen easily. This further validates the effectiveness of our proposed T-DCS in measuring the relative defense strength of different types of attacks. (2) The differences in DCS scores between splitVFL and aggVFL settings (in Section 6) are also consistent with the human domain knowledge that splitVFL is less vunerable to attacks compared to aggVFL, which is consistent with findings in [1], demonstrating the effectiveness of the metrics.
> - **Alignment with human intuition.** By definition (Eq. (1)), our DCS score measures its closeness to a perfect defense (Fig. 2).  Intuitively, a higher DCS score represents a defense that achieves a lower AP at a higher level of MP compared to others. This definition aligns with the defense target, i.e. limiting the AP to a lower level while maintaining a higher level of MP.
>
> **Reference**
>
> [1] Yan Kang, et al. "A framework for evaluating privacy-utility trade-off in vertical federated learning". arXiv preprint arXiv:2209.03885, 2022

---

> > ### Author Response · Authors · 2023-11-18
> > **Responses to Questions and Weaknesses from R-1nCt (2)**
> >
> > ### **Response to Weakness 3**
> >
> > Thanks for the suggestion. We further demonstrate the workflow of VFLAIR in Appendix D. We also post Fig. 10 in Appendix D here ([this figure (link)](https://anonymous.4open.science/r/VFLAIR_discussion/workflow_new.png)) for direct access  .
> > In summary, VFLAIR contains 4 key modules, namely Config Module, Load Module, Train \& Evaluate Module and Metrics Module.
> >
> > In the workflow, the Config Module first processes the user-specified configurations, then passes it to the Load Module for party preparation. Each party separately loads its dataset and local model in the Load Module. Afterwards, the training pipeline starts in the Train \& Evaluate Module. Communication protocols and all the supported defense (all are training time defense) are implemented and integrated into the training pipeline. Training-time attacks (e.g. TB and NTB attacks) are integrated and evaluated within the training pipeline while inference-time attacks that do not affect the training flow (e.g. LI and FR attacks) are launched and evaluated after the training pipeline. Finally, the Metrics Module processes the recorded results and produces evaluation metrics including MP, AP, communication rounds (\#Rounds), amount of information transmitted between parties each round (Amount), execution time (EXec.Time), DCS, T-DCS and C-DCS.
> >
> > ### **Response to Weakness 4**
> >
> > Thank you for your constructive suggestions. We have reordered the results and discussions systematically in Section 6 in our paper, which now exhibits the results according to model partition, communication protocols, encryption, number of participants, model architectures, attacks and defenses etc. We have also summarized and highlighted insights from our findings at the beginning of each evaluation section. Please refer to our modified manuscript for details.
> >
> > ### **Response to Weakness 5**
> > Thanks for pointing this out. We have removed the bullet point in our Related Work section according to your advice.

---

> ### Author Response · Authors · 2023-11-22
> **We are anxiously looking forward to your replies to our responses**
>
> Dear reviewer, do our responses to your questions as well as the updated paper address your concerns? We have added a workflow figure in (in the appendix and the link in the response), testified the effectiveness of our DCS evaluation (in the response and in the appendix) and reorganized the result section (Section 6) and related work section (Section 2) according to your reviews. We hope the above have solved your concerns and we are still looking forward to your further replies. We will remain open and available if there is any further concerns.

---

> ### Author Response · Authors · 2023-11-23
> **We are still anxiously looking forward to your replies to our responses**
>
> Dear reviewer, we are still anxiously looking forward to your replies to our responses. Time is limited with less than **5** hours till the end of the discussion period. We are curious to know if our responses to your questions as well as the updated paper have already addressed your concerns. We have put a lot of effort into working to address your concerns and to improve our work. We summarize our modifications with respect to your reviews as below:
>
> - We testified the effectiveness of our DCS evaluation in our response from 3 different aspects to answer your question, including Robustness, Sensitivity and Alignment with human intuition. We have also included these analyses in our updated paper in Section 6 and Appendix I.2.6.
> - We have added a workflow figure in (in appendix D and a link for quick access in the response).
> - We reorganized the result section (Section 6) in a more systematic way according to your reviews. We also modified the related work section (Section 2) according to your helpful suggestion.
> - We carefully considered your comment on the paper title and clearly presented the reasons why we choose to kept it unchanged.
>
> We hope the above have solved your concerns and we are still looking forward to your further replies. We will remain open and available if there is any further concerns.

---

### Official Review · Reviewer_z5tx · 2023-11-01

**Soundness:** 3 good
**Presentation:** 3 good
**Contribution:** 3 good
**Rating:** 8
**Confidence:** 4

**Summary:**

The paper presents a research library and benchmark named VFLAIR for vertical federated learning. VFLAIR contains 9 datasets, 29 models, 2 communication protocols, 3 data partitioning, 11 attacks, and 8 defense methods. Model performance, attack and defense performance, and communication protocol comparison are comprehensively evaluated.

**Strengths:**

1. While vertical federated learning is a promising research direction with many real-world applications, it is less exploited compared with horizontal federated learning. Unlike horizontal federated learning systems, there is a lack of a comprehensive vertical FL library. This work is a significant contribution to the FL community.

2. The library is comprehensive and includes many models, attacks, and defense methods.

3. Experiments are extensive, especially for the attack and defense part.

**Weaknesses:**

1. The writing needs to be further improved. The paper claims that VFLAIR is a lightweight and extensible framework but does not demonstrate why it is. The introduction of the framework is limited. Besides introducing the components of VFLAIR in Figure 1, the paper should also introduce what is the systematic design of VFLAIR and demonstrate why it is very easy to use and extend.

2. The insights in Section 6 should be highlighted. Currently, the paragraph is too long (especially Section 6.2) and readers are hard to find interesting results from the experiments. I suggest the authors put the insights at the beginning of each subsection.

3. It seems that the paper divides the datasets into multiple subsets equally. Non-IID data partitioning is an important factor in FL and would be good to be included in VFLAIR. Also, it’d be better to include real-world vertical federated datasets besides partitioning a centralized dataset.

4. The communication protocols in the library are not rich. Only two methods are considered.

**Questions:**

1. Can you demonstrate how to use VFLIR and how to extend it?

2. Can you summarize and highlight interesting findings at the beginning of each subsection of Section 6.1?

3. Will you consider including more communication protocols, real-world vertical federated datasets, and data partitioning methods in the library?

---

> ### Author Response · Authors · 2023-11-18
> **Responses to Questions and Weaknesses from R-z5tx**
>
> ### **Response to Weakness 1 and Question 1**
>
> In order to better illustrate the systematic design of VFLAIR, we add Fig. 10 in Appendix D (see also [this figure (link)](https://anonymous.4open.science/r/VFLAIR_discussion/workflow_new.png)) to demonstrate the key modules and workflow of VFLAIR. We also add a step by step guidance for using and extending VFLAIR in Appendix C. In summary, building on a modularized design of components covering all essential aspects of a VFL system, using VFLAIR only requires 5 easy steps as shown in Fig. 8 in Appendix C (see also [this figure (link)](https://anonymous.4open.science/r/VFLAIR_discussion/demo/demo_how_to_use.png)), with a single line to run: "python main_pipeline.py --configs my_config".
>
> For extending VFLAIR, we summarized the necessary steps and also included 2 demo cases ("how to add a new attack" and "how to add a new local model") in Appendix C. In summary, extending VFLAIR requires only modifying the configuration file, implementing code by inheriting corresponding classes (e.g. attacker) or by adding the model architecture and run.
>
> Please see Appendix C and D for more details.
>
>
> ### **Response to Weakness 2 and Question 2**
> Thanks for the suggestions. We have added and highlighted the summary of insights at the beginning of each paragraph throughout section 6. Please see our modified manuscript.
>
>
> ### **Response to Weakness 3 and Question 3**
> Thanks a lot for your suggestions. We first need to clarify that, different from horizontal federated learning (HFL) where datasets are partitioned by samples, "non-IID data" distribution is not considered in VFL because datasets are partitioned by features. That said, we did consider "unequal split of features among each party" in our experiments with justifiable partitioning methods. For example, as we already included in our submission, NUSWIDE is a cross-modal dataset which contains an image field of 634 features and a text field of 1000 features, which are naturally partitioned into two parties by modality, resulting in an unbalanced feature set among parties. This also aligns with real-world data distribution where text and its paired image are held by different parties.
>
> Despite the rapid growth of real-world VFL applications, it is still difficult to collect truly distributed real-world dataset due to data privacy and regulations. However, we have additionally added Criteo, Avazu, Cora and News20 into our benchmark results in Tab. 7 in Section 6.1 (and below). They are considered real-world datasets for typical VFL applications (Criteo and Avazu for click through rate prediction in advertising, Cora for node classification in citation network, and News20 for classification of news data). As shown in the table below, we adopt partition strategies aligning with human observation of features, resulting in unbalanced features among parties (see also Tab.10 in Appendix H for complete dataset and partition details).
>
> (1) Tab. 7 (in paper). MP, communication rounds (\#Rounds) for reaching specified MP with 4 real-world datasets of NN-based VFL with **FedSGD** communication protocol.
> |  Dataset  |      aggVFL     |  aggVFL  |     splitVFL    | splitVFL |
> |:---------:|:---------------:|:--------:|:---------------:|:--------:|
> |            |        MP       | \#Rounds |        MP       | \#Rounds |
> |   Criteo  | 0.715$\pm$0.053 |     2    | 0.744$\pm$0.001 |     3    |
> |   Avazu   | 0.832$\pm$0.001 |     5    | 0.832$\pm$0.001 |     9    |
> |    Cora   | 0.721$\pm$0.004 |    11    | 0.724$\pm$0.012 |    13    |
> | News20-S5 | 0.882$\pm$0.014 |    57    | 0.893$\pm$0.013 |    61    |
>
> (2) Feature (dataset) partition strategies that are unequal. (p): passive party, (a): active party for column 'Number of Features'.
> | Dataset | Number of Features  |                Partition Criteria                |       Added during Discussion      |
> |:-------:|:------------------------------------------:|:------------------------------------------------:|:----------------------------------:|
> | NUSWIDE |              1000 (p), 634 (a)             |           feature type (text or image)           | No (already included in subission) |
> |  Credit |                12 (p), 11(a)               | feature meaning (bill/previous statement or not) | No (already included in subission) |
> |  Criteo |               13 (p), 26 (a)               |         feature type (discrete or dense)         |                 Yes                |
> |  Avazu  |                9 (p), 13 (a)               |          feature type (anonymous or not)         |                 Yes                |

---

> > ### Author Response · Authors · 2023-11-18
> > **Responses to Questions and Weaknesses from R-z5tx (2)**
> >
> > ### **Response to Weakness 4 and Question 3**
> >
> > We have added 3 new communication protocols, namely Quantizate [1], Top-k [1] and CELU-VFL [2] into our benchmark esperiments with the results shown in Tab. 6 in Section 6.1. We also plan to include more communication protocols  in the future. We also place the results here for easier access.
> >
> > (3) Tab. 6 (in paper). MP, communication rounds (\#Rounds), amount of information exchanged per round (Amount) and total amount of exchanged information (Total) with different communication protocols of NN-based VFL under **aggVFL** setting. Q=5 when **FedBCD** and **CELU-VFL** are used, otherwise Q=1. For **Quantize**, b=16 while for **Top-k**, top 90% of elements are kept in forward local model prediction. 'Total' column is the total amount that equals to \#Rounds x Amount.
> > |          | MNIST           | MNIST   | MNIST      |  MNIST    | NUSWIDE         | NUSWIDE |  NUSWIDE   | NUSWIDE   |
> > | -------- | --------------- | ------- | ---------- | --------- | --------------- | ------- | ---------- | --------- |
> > |          | MP              | #Rounds | Amount(MB) | Total(MB) | MP              | #Rounds | Amount(MB) | Total(MB) |
> > | FedSGD   | 0.972$\pm$0.001 | 150     | 0.156      | 23.438    | 0.887$\pm$0.001 | 60      | 0.039      | 2.344     |
> > | FedBCD   | 0.971$\pm$0.001 | 113     | 0.156      | 17.656    | 0.882$\pm$0.001 | 26      | 0.039      | 1.016     |
> > | Quantize | 0.959$\pm$0.006 | 161     | 0.117      | 18.867    | 0.881$\pm$0.002 | 94      | 0.029      | 2.754     |
> > | Top-k    | 0.968$\pm$0.001 | 150     | 0.148      | 22.266    | 0.887$\pm$0.001 | 60      | 0.037      | 2.227     |
> > | CELU-VFL | 0.971$\pm$0.002 | 105     | 0.156      | 16.406    | 0.880$\pm$0.001 | 25      | 0.039      | 0.977     |
> >
> > **Reference**
> >
> > [1] Castiglia, Timothy J., et al. "Compressed-vfl: Communication-efficient learning with vertically partitioned data." International Conference on Machine Learning. PMLR, 2022.
> >
> > [2] Fu, Fangcheng, et al. "Towards communication-efficient vertical federated learning training via cache-enabled local updates." arXiv preprint arXiv:2207.14628 (2022).

---

> ### Author Response · Authors · 2023-11-22
> **We are anxiously looking forward to your replies to our responses**
>
> Dear reviewer, do our responses to your questions as well as the updated paper address your concerns? We have added more unequal dataset partition methods (in Tab. 7 in Section 6); more communication protocols (in Tab. 6 in Section 6) like quantization, Top-k compression and CELU-VFL to our VFLAIR platform and benchmark experiments according to your reviews. We have also added a step by step user guidance as well as 2 demos on how to extend VFLAIR in the appendix (Appendix C). Moreover, we have reorganized Section 6 and highlighted the insights of each paragraph in our updated paper. We hope the above have solved your concerns and we are still looking forward to your further replies. We will remain open and available if there is any further concerns.

---

> ### Comment · Reviewer_z5tx · 2023-11-22
>
> Thanks for your response. The revision has addressed most of my concerns. The paper is more clear now. I have raised my score to 8. For the benchmark, it'd be great if the authors could make it a Python library so that researchers can easily import the approaches for more flexible usage.

---

> > ### Author Response · Authors · 2023-11-23
> > **Thanks for your further comments**
> >
> > We are really happy to hear that we have addressed most of your concerns! Thanks for your further advice on making VFLAIR a Python library. We will add this to our future plane for further facilitating future researches. We promise.

---

### Author Response · Authors · 2023-11-18
**To all Reviewers, AC and PC: Contribution and summary of modifications during discussion**

We deeply thank the reviewers for their appreciation of this work and insightful feedbacks. Your suggestions greatly help us to improve and enrich our work. In the following discussions, we individually address your comments.

We would like to clarify first that our contribution lies not only on the 11 attack and 8 defense we evaluated with our proposed DCS metrics, but also lies in the richness of our framework as it covers various aspects of VFL including 13 datasets (4 added during discussion) of different modality and data partition methods, 29 model architectures including tree, LR, MLP, CNN, GCN, LSTM etc., 2 model partition methods, 5 communication protocols (3 added during discussion) and 1 encryption technique. We also perform extensive experiments to benchmark model utility, communication and computation efficiency under various afformentioned settings.

According to all the feedback from our reviewers, we mainly modified our manuscript as listed below:
- 4 real-world datasets are added, including real-world financial datasets like Criteo and Avazu, and graph data like Cora which utilize GCN as local models. More unbalanced dataset partitioning settings are considered in addition to those that are already included.
- 3 new communication protocols are added and compared (on model utility and communication cost).
- The result section (Section 6) is reorganized systematically according to different settings. We also highlight interesting insights at the beginning of each paragraph throughout Section 6. Due to space limitations, we move the original Fig. 6 to the appendix (Fig. 16 currently).
- Step-by-step guidance on how to use and extend our VFLAIR framework is additionally provided in Appendix C with 2 demos.
-  A Workflow of VFL is also demonstrated with Fig. 10 in Appendix D.

We hope our revised manuscript and additional experimental results can help address our reviewers' concerns and doubts.

---

> ### Author Response · Authors · 2023-11-20
> **To all Reviewers, AC and PC: Highlight of modification available**
>
> Dear reviewers, AC and PC:
>
> We have updated our paper and supplementary material for adding highlights on the modifications we made to our paper. We look forward to your further replies. Thanks.

---

### Meta-Review · Area_Chair_doaL · 2023-12-20

**Metareview:**

Reviewers appreciate the efforts that the authors put into making a research library for an important problem. There are concerns from the reviewers about the scoping and be more precise about the messaging and the title -- we hope that the authors can take them into consideration in the next version of the paper.

**Justification For Why Not Higher Score:**

There are rooms for improvement on presentation and research contributions beyond a research library.

**Justification For Why Not Lower Score:**

Great contribution to a timely problem.

---

### Decision · Program_Chairs · 2024-01-16

Accept (poster)